# Glycoproteomic landscape and structural dynamics of TIM family immune checkpoints enabled by mucinase SmE

Joann Chongsaritsinsuk[1,10], Alexandra D. Steigmeyer [1,10], Keira E. Mahoney [1,10], Mia A. Rosenfeld [2,10], Taryn M. Lucas [1], Courtney M. Smith[3], Alice Li[3], Deniz Ince[1], Fiona L. Kearns[2], Alexandria S. Battison [1], Marie A. Hollenhorst[4,5,6], D. Judy Shon [4], Katherine H. Tiemeyer [4], Victor Attah[1], Catherine Kwon[1], Carolyn R. Bertozzi [4,7], Michael J. Ferracane [8], Mark A. Lemmon [3], Rommie E. Amaro [2,9] & Stacy A. Malaker [1]✉

Mucin-domain glycoproteins are densely O-glycosylated and play critical roles in a host of biological functions. In particular, the T cell immunoglobulin and mucin-domain containing family of proteins (TIM-1, -3, -4) decorate immune cells and act as key regulators in cellular immunity. However, their dense O-glycosylation remains enigmatic, primarily due to the challenges associated with studying mucin domains. Here, we demonstrate that the mucinase SmE has a unique ability to cleave at residues bearing very complex glycans. SmE enables improved mass spectrometric analysis of several mucins, including the entire TIM family. With this information in-hand, we perform molecular dynamics (MD) simulations of TIM-3 and -4 to understand how glycosylation affects structural features of these proteins. Finally, we use these models to investigate the functional relevance of glycosylation for TIM-3 function and ligand binding. Overall, we present a powerful workflow to better understand the detailed molecular structures and functions of the mucinome.

Mucin-domain glycoproteins are characterized by dense O-glycosylation that contributes to a unique, bottle-brush secondary structure that can extend away from the cell surface or form extra-cellular gel-like secretions[1,2]. Mucin-type O-glycans are characterized by an initiating α-*N*-acetylgalactosamine (GalNAc) that can be further elaborated into several core structures containing sialic acid, fucose, and/or ABO blood group antigens. As a result, mucin domains serve as highly heterogeneous stretches of glycosylation that exert both

biophysical and biochemical influences on the cellular milieu[3,4]. The canonical family of mucins, e.g. MUC2 and MUC16, bear massive mucin domains and are heavily implicated in various diseases[5,6]. That said, many other proteins contain mucin domains that do not reach that size or complexity but are nonetheless functionally important. Indeed, we recently introduced the human "mucinome", which comprises hundreds of proteins thought to contain the dense O-glycosylation that is characteristic of mucin domains[7]. For

[1]Department of Chemistry, Yale University, New Haven, CT 06511, USA. [2]Department of Chemistry and Biochemistry, University of California, San Diego, La Jolla, CA 92093, USA. [3]Yale Cancer Biology Institute and Department of Pharmacology, Yale University School of Medicine, New Haven, CT 06520, USA. [4]Department of Chemistry and Sarafan ChEM-H, Stanford University, Stanford, CA 94305, USA. [5]Department of Pathology, Stanford University, Stanford, CA 94305, USA. [6]Department of Medicine, Division of Hematology, Stanford University, Stanford, CA 94305, USA. [7]Howard Hughes Medical Institute, Stanford University, Stanford, CA 94305, USA. [8]Department of Chemistry, University of Redlands, Redlands, CA 92373, USA. [9]Glycobiology Research and Training Center, University of California, San Diego, La Jolla, CA 92093, USA. [10]These authors contributed equally: Joann Chongsaritsinsuk, Alexandra D. Steigmeyer, Keira E. Mahoney, Mia A. Rosenfeld ✉e-mail: stacy.malaker@yale.edu

instance, platelet glycoprotein 1bα (GP1bα) interacts with Von Willebrand Factor to mediate platelet adhesion, and mutations in GP1bα are involved in platelet-type Von Willebrand disease[8]. C1 esterase inhibitor (C1-Inh) is a serine protease inhibitor, and a deficiency of this protein is associated with hereditary angioedema[9]. Finally, the T cell immunoglobulin and mucin domain-containing protein family (TIM-1, TIM-3 and TIM-4) are critical regulators of cellular immune responses and have substantial importance in immune-oncology[10,11]. Although considerable progress has been made in the biological and analytical analyses of these and other mucin-domain glycoproteins, much remains unknown regarding their glycan structures, glycosylation site-specificity, and functional roles within the cellular environment.

This gap in knowledge is due, in part, to the challenges associated with studying mucins by mass spectrometry (MS)[12–14]. Mucins present unique challenges at each stage of a typical MS workflow. One of the most well-documented issues is the resistance of densely O-glycosylated domains to trypsin digestion[2,15,16]. To address this challenge, several proteases have been introduced that selectively cleave at or near O-glycosylated residues thereby revolutionizing the field of O-glycoproteomics[17–20]. These enzymes are aptly named O-glycoproteases; those that prefer mucin-domain glycoproteins are often termed mucinases. The first of these enzymes, OgpA, was characterized as an O-glycoprotease that cleaves N-terminally to glycosylated Ser or Thr residues but is hindered by the presence of sialic acid[19,21]. We introduced StcE as a mucinase that selectively digests mucin domains with a cleavage motif of T/S*_X_T/S, wherein the asterisk indicates a mandatory glycosylation site; we subsequently developed a mucinase toolkit that displays a wide range of cleavage specificities[17,18]. More recently, ImpA was commercialized and, like OgpA, cleaves N-terminally to glycosylated Ser or Thr residues but, unlike OgpA, is less restricted by the glycans present[22]. While these enzymes have aided in the analysis of many O-glycoproteins and mucins, the use of each enzyme is accompanied by drawbacks. An ideal, broad-specificity O-glycoprotease conducive to MS analysis has not yet been characterized.

Another issue surrounding the characterization of mucin-domain glycoproteins is that typical structural biology techniques are not well suited for glycoproteins, let alone the dense glycosylation characteristic of mucin domains. As covered in our recent review, current knowledge regarding mucin secondary structure originates from various low-resolution images generated by atomic force microscopy (AFM), scanning electron microscopy (SEM), and cryogenic electron microscopy (cryoEM)[2,23–25]. While this has allowed us to definitively visualize the linearity of mucin protein backbones, by nature of the techniques, we are unable to (a) discern the individual glycans and how they contribute to changes in protein structure or (b) observe the mucin protein dynamics[2]. More recently, many advances have been made in computational modeling of glycoproteins[26]. Molecular dynamics (MD) simulations have revealed some of the many roles that glycans play in the structure, stability, dynamics, and function of glycoproteins. Most notably, Amaro and colleagues revealed the functional role of the glycan shield in the activation mechanism of the SARS-CoV-2 spike protein[27,28]. That said, while mucins have been subjected to MD simulations previously, they are often modeled with a static glycan structure, lack precise glycosylation information, and are simulated with coarse-grained MD. Taken together, for mucin-domain glycoproteins, we generally do not know the glycoproteomic landscape nor how the glycans work in concert to control protein and cellular dynamics.

Here, we present a powerful technique to map the complex glycosylation within mucin domains and pair this information with MD simulations in order to better understand how glycans affect glycoprotein secondary structure and dynamics. We first introduce a mucinase, Serratia marcescens Enhancin (SmE), and demonstrate its

unique ability to cleave at glycosites decorated by a myriad of glycans enabled enhanced glycosite and glycoform analysis by MS. With SmE in-hand, we then obtained complete O-glycoproteomic information for all TIM family proteins and demonstrated that TIM-3 has markedly fewer O-glycosites when compared to TIM-1 and -4. To better understand how these glycans affect overall protein structure, we then employed MD simulations and showed that TIM-3 has a much shorter persistence length and higher flexibility than TIM-4. Finally, we demonstrated lattice formation induced by galectin-9 (Gal-9) binding glycans on TIM-3 enhanced binding affinity/avidity for membranes containing PtdSer. Overall, this workflow aids in unraveling the complex molecular mechanisms behind mucin domains, their glycan patterns, and their contribution to cellular biology.

## Results
### Characterization of SmE cleavage motif and glycan specificity
Various microorganisms found within mucosal environments secrete proteolytic enzymes that have been shown to be advantageous tools for MS analysis of mucins. We and others have mined the microbiota to generate a toolkit of O-glycoproteases, each with unique peptide and glycan specificities[17,18,20,22]. In particular, Serratia marcescens is a pervasive opportunistic pathogen in humans. This organism secretes a mucinase, SmE, that is a viral enhancin protein shown to promote arboviral infection of mosquitoes by degrading gut membrane-bound mucins[29]. SmE contains a catalytic domain belonging to the Pfam family PF13402 (peptidase M60, enhancin, and enhancin-like or M60-like family) that is defined by a conserved HEXXH metallopeptidase motif[30]. We expressed SmE as a 94-kDa soluble, His-tagged protein in E. coli at a high-yield expression level of 65 mg/L (Supplementary Fig. 1). To determine SmE's mucin selectivity, we digested glycoproteins with and without mucin domains. SmE preferentially cleaved proteins with mucin domains—C1-Inh, CD43, and TIM-1—whereas it did not significantly cleave the non-mucin glycoproteins fibronectin and fetuin (Supplementary Fig. 2).

As in our previous work, we characterized SmE's cleavage motif using biologically relevant mucin-domain glycoproteins (C1-Inh, TIM-1, TIM-3, TIM-4, and GP1bα). These proteins were digested with SmE and subjected to MS analysis (Fig. 1A). Manually validated glycopeptides were mapped to protein sequences to identify sequence windows, which allowed the determination of minimum sequence motifs. As demonstrated in Fig. 1B, SmE cleaved N-terminally to a glycosylated Ser or Thr residue. SmE accommodated a variety of O-linked glycans at the P1' position (pie chart, right), including sialylated core 1 and core 2 O-linked glycans as well as fucosylated ABO blood group antigens. SmE was also able to accommodate O-glycosylation at the P1 position (pie chart, left). Given the higher percentage of smaller O-glycan structures (GalNAc, GalNAc-Gal), it appears that the P1 O-glycosylation tolerance was less permissive at this position. However, these pie charts represent only site-localized glycan structures; site-localization at the C-terminus of the peptide is more challenging due to the lack of positive charge. Thus, the apparent preference for smaller glycan structures at the P1 position is likely due to issues in glycoproteomic analysis as opposed to an inability of SmE to cleave at residues bearing larger glycans.

SmE provides a complementary cleavage profile to StcE which, as mentioned above, cleaves at a T/S*_X_T/S motif. In that work, we also demonstrated that StcE is non-toxic to cells and can be employed to release mucins from the cell surface[17]. Many researchers have since used our mucinase toolkit, especially StcE, to remove mucins from the cell surface and/or degrade mucins in biological samples[1,31–34]. Importantly, the MUC1 repeat sequence HGVTSAPDTRPAPGSTAPPA does not contain StcE's cleavage motif, so limited digestion occurs within this region[35]. Given that SmE has a complementary cleavage motif, a combinatorial treatment strategy could enable further degradation of mucins from various biological samples. Thus, we sought to

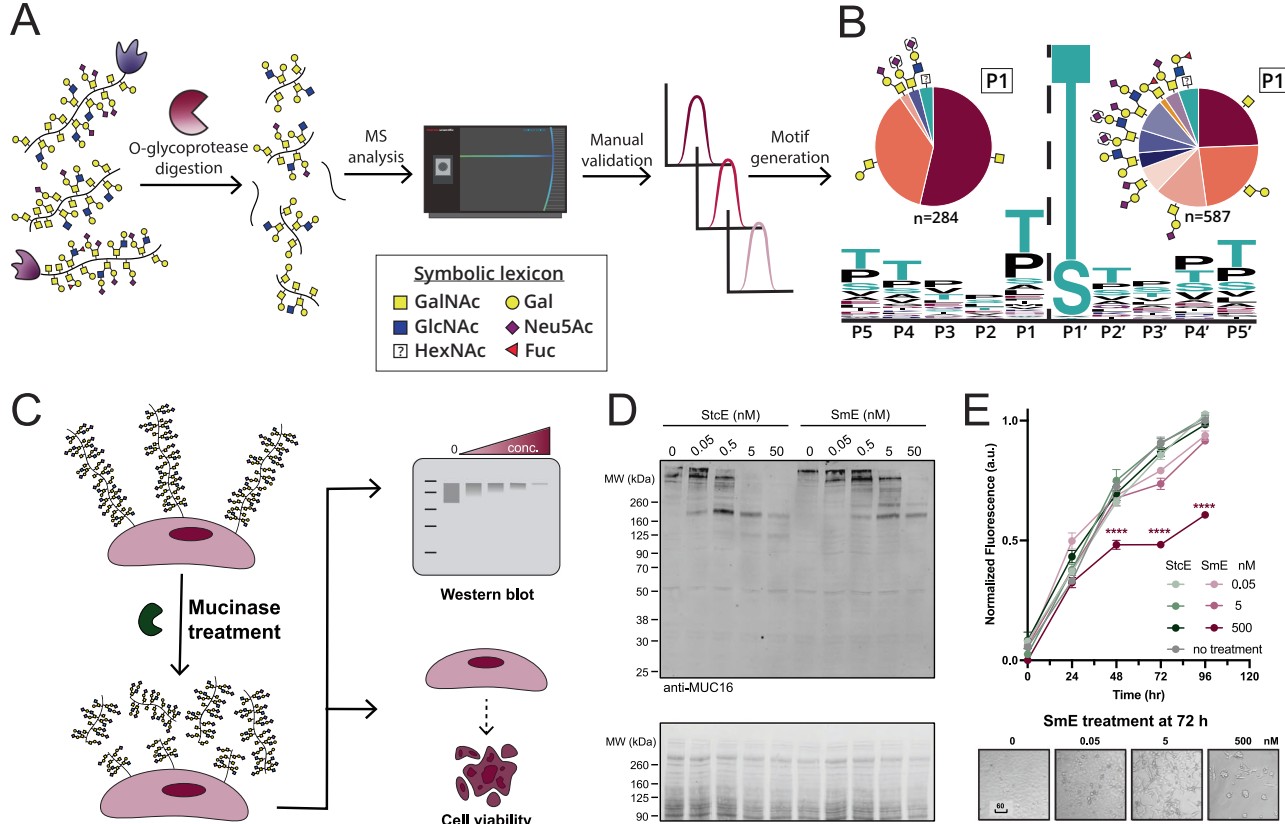

**Fig. 1 | Characterization of mucinase SmE for analysis and degradation of mucin-domain glycoproteins. A** Workflow for generating consensus sequence of SmE. **B** Five recombinant mucin-domain glycoproteins were digested with SmE and subjected to MS analysis. Peptides present in the mucinase-treated samples were used as input for weblogo.berkeley.edu (±5 residues from the site of cleavage). Parentheses around sialic acids (purple diamond) indicate that its linkage site was ambiguous. **C** Workflow to evaluate the toxicity and cell surface activity of SmE. **D** HeLa cells were treated with StcE (left) or SmE (right) at the noted concentrations for 60 min. Following treatment, the cells were lysed in 1X NuPAGE LDS Sample Buffer with 25 mM DTT, subjected to separation by gel electrophoresis, and probed for MUC16 by Western blot. Proteins were transferred to a nitrocellulose membrane using the Trans-Blot Turbo Transfer System (Bio-Rad) at a constant 2.5 A for 15 min.

Total protein was quantified using REVERT stain before primary antibody incubation overnight at 4 °C. An IR800 dye-labeled secondary antibody was used according to the manufacturer's instructions for visualization on a LiCOR Odyssey instrument. **E** HeLa cells were treated with SmE and StcE at 0, 0.05, 5, and 500 nM. At t = 0, 24, 48, 72, 96 h post-treatment, PrestoBlue was added according to manufacturer's instructions. After 2 h, the supernatant was transferred to a black 96 well plate and analyzed on a SPECTRAmax GEMINI spectrofluorometer using an excitation wavelength of 544 nm and an emission wavelength of 585 nm. Data are presented as mean values ± SD for $n = 3$ biologically independent samples. Statistical significance was determined using the two-way ANOVA analysis in Graphpad PRISM software and is reported with respect to the 'no treatment' control condition. ***$p = 0.0001$, ****$p < 0.0001$. Source data are provided in the Source Data file.

understand whether SmE is similarly capable of digesting mucins from the cell surface, and whether the enzyme is likewise non-toxic to cells (Fig. 1C). Treating HeLa cells with SmE resulted in a reduction in MUC16 staining by Western blot in a manner comparable to that of StcE (Fig. 1D). We also detected released MUC16 fragments in the supernatant of SmE treated cells (Supplementary Fig. 3). Importantly, SmE was not toxic to cells under conditions used previously for StcE, although cell death was observed at higher SmE concentrations over longer treatment durations (Fig. 1E and Supplementary Fig. 4). Taken together, these results indicate that SmE, like StcE, can effectively cleave mucins from the cell surface and serve as a tool to probe mucin biological function.

### SmE outperforms commercial O-glycoproteases OgpA and ImpA for mucin analysis

To compare the activity of SmE in context with widely used, commercially available O-glycoproteases, we decided to benchmark against both OgpA and ImpA[19,22,36,37]. OgpA was originally identified in *Akkermansia muciniphila*, a commensal bacterium known to regulate mucin barriers through controlled degradation[38]. As mentioned above, OgpA is reported to cleave N-terminally to O-glycosylated Ser or Thr residues, with highest affinity towards asialylated core 1 species[39,40]. Thus, typical

workflows with this enzyme involve removal of sialic acids, which limits its use to site-mapping rather than providing information on native glycan structures. ImpA is derived from *Pseudomonas aeruginosa*, an opportunistic bacterial pathogen that can cause severe infection[41]. Like OgpA, ImpA cleaves N-terminally to an O-glycosylated Ser or Thr residue; however, this enzyme has been reported to accommodate more complex, sialylated glycans, expanding its glycoproteomic potential beyond that of OgpA. That said, it has been observed that cleavage by ImpA is influenced by amino acid identity in the P1 position[22,37].

To directly compare the activities of OgpA, ImpA, and SmE, mucin-domain glycoproteins were digested in the presence and absence of sialidase, followed by gel electrophoresis (Supplementary Fig. 5), MS analysis, and manual glycopeptide validation (Supplementary Data 1–5). Additionally, we included fetuin to investigate the enzymes' selectivity for mucin glycoproteins (Supplementary Data 6). As demonstrated in Fig. 2A, B, we confirmed the reported cleavage motifs and glycan preferences of both OgpA and ImpA. Notably, after ImpA digestion, we did not detect any glycosites in the P1 position, suggesting that ImpA does not cleave between two glycosylated residues. Given that mucin domains contain many neighboring O-glycosites, this presents a significant limitation in the use of ImpA for mucinomic analysis.

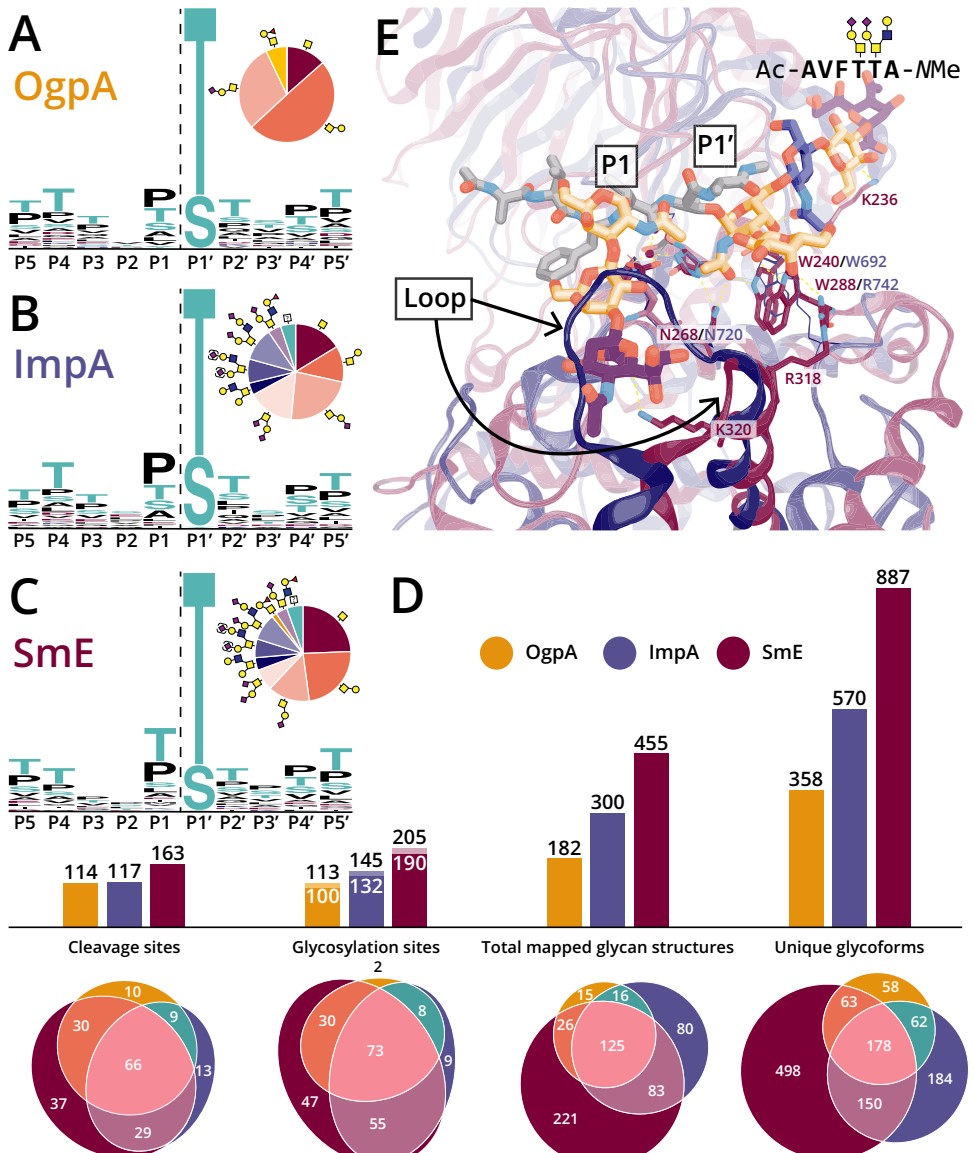

**Fig. 2 | SmE outperformed commercial O-glycoproteases due to its structural permissiveness.** Cleavage motifs for (**A**) OgpA, (**B**) ImpA, and (**C**) SmE as determined by digestion followed by MS and manual curation of glycopeptides. **D** Bar graphs and Euler plots demonstrating counts and overlap between enzymes regarding the number of observed cleavage sites, localized glycosylation sites, total glycan structures, and unique glycoforms. For the "Glycosylation sites" bar graph, glycosites localized via MS are denoted by white numbers; black numbers above include implied glycosites where cleavage was observed but the glycosite was not localized. **E** A glycopeptide docked in the active site of SmE (maroon) and ImpA (blue), highlighting differences between key loops and residues of the two O-glycoproteases.

In contrast, we found that SmE activity was not limited by glycan complexity or adjacent glycosylation (Figs. 1B, 2C). Perhaps for this reason, digestion with SmE greatly improved the depth and coverage of the glycoproteomic landscape for each mucin-domain glycoprotein we investigated. The number of cleavage sites, unique O-glycosites, mapped glycan structures, and total glycoforms were all determined (Fig. 2D, see Supplementary Fig. 6 for maps of all cleavage events). Here, total mapped glycan structures were calculated by counting every O-glycan associated with each O-glycosite. Unique glycoforms refers to the total number of validated glycopeptides that were identified from protein digestions. In our analyses, we found that OgpA allowed for the identification of 113 glycosites, 182 mapped glycan structures, and 358 unique glycoforms. ImpA demonstrated significant improvement over OgpA, enabling localization of 145 glycosites, 300 mapped structures, and 570 glycoforms. Even more impressively, SmE digestion allowed us to identify 205 glycosites, 455 mapped structures, and 887 glycoforms.

Notably, using SmE permitted the identification of 47 unique glycosites, 221 glycan structures, and 498 glycoforms that were not detected using the other enzymes. Previously, glycomic and glycoproteomic analyses of C1-Inh hinted at a total of approximately 25 O-glycosites; however, these were sparingly localized to individual residues[9,42,43]. SmE enabled full glycoproteomic mapping of the C1-Inh mucin domain (Supplementary Fig. 7, Supplementary Data 1), thus reinforcing the utility of this enzyme. Taken together, SmE greatly outperformed both OgpA and ImpA with regard to glycoproteomic analysis of mucin domains. For a discussion on limitations associated with SmE for MS analysis, please see the Supplemental Information (Supplementary Figs. 8–10).

### Molecular modeling helps rationalize different substrate selectivity between SmE and ImpA

Previously, we used molecular docking to better understand StcE's substrate selectivity[17]. Given that SmE and ImpA have catalytic domains

belonging to the same Pfam family[44,45], yet have quite different cleavage motifs, we decided to again use molecular modeling to understand the structural basis behind these differences. OgpA is a single-domain enzyme with a catalytic metzincin motif that is not defined by Pfam and is more distantly related to SmE and ImpA, and therefore was excluded from these comparison analyses.

To date, four unique crystal structures of ImpA have been determined[30,46], including one with a ligand bound at the active site and a second with a ligand bound at an exosite located in the N-terminal domain (PF18650). The structure of SmE, on the other hand, has not yet been elucidated. As such, we aligned structures of all characterized enzymes with a PF13402 catalytic domain[47–50], including the SmE structure recently predicted by AlphaFold (Supplementary Fig. 11)[51,52]. We then docked a TIM-4-based bisglycosylated peptide into the predicted SmE structure to provide insight into potential substrate recognition.

In its cocrystal structure with (Gal-GalNAc)Ser[30], ImpA uses specific residues to recognize glycans branching from P1' (Fig. 2E, blue). The side chains of the conserved residues Trp692 and Asn720 form polar contacts with the carbonyl of the GalNAc moiety, and the side chain of Arg742 also interacts with the GalNAc. The Gal moiety, on the other hand, does not interact with the enzyme and is projected into solvent. The helix lining the active site of ImpA is short[36], indicating that branched glycans could reasonably be accommodated by this enzyme in a similar manner to that seen with ZmpB and ZmpC[30]. These combined factors likely impart ImpA with activity on substrates bearing mucin-like glycosylation at P1' but little selectivity for particular modifications beyond the initiating GalNAc moiety.

In our docked structure of SmE with a bisglycosylated TIM-4-based glycopeptide (Fig. 2E, red), we found analogous interactions between conserved residues Trp240 (replacing Trp692 in ImpA) and Asn268 (replacing Asn720 in ImpA) and the GalNAc initiating from P1'. Additional unique contacts were found between the 3-OH of GalNAc and the side chain of Trp288, rather than with an Arg residue as seen in crystal structures of ImpA and other PF13402-containing enzymes. Although there is an Arg residue (Arg282) nearby in the sequence, it is not predicted to flank the GalNAc moiety in the AlphaFold structure, and we found that this residue is less conserved in PF13402-containing enzymes than previously suggested (Supplementary Fig. 11)[47]. Interestingly, a different Arg side chain (Arg318) contacted the 4-OH of the Gal residue, likely imparting further specificity for mucin-like glycosylation. Similar to ImpA, SmE is predicted to have a short active site helix. Here, the branched glycan was well accommodated by the enzyme in our model, and we commonly found orientations that formed contacts between the ligand and the active site helix as well as the preceding loop, similar to what was observed between ZmpB/ZmpC and their branched ligands (Supplementary Fig. 12)[30,48]. Together, these results suggest that SmE has better recognition of the initiating GalNAc and Gal residues, and that it likely forms additional interactions with branched glycans.

Neither the ImpA crystal structure nor the SmE predicted structure contain a beta hairpin analogous to the one found to recognize P1 glycans by the mucinase AM0627 (Supplementary Figs. 11, 13), which also allows glycosylated P1 Ser/Thr residues[50]. Thus, the steric environment in this region is primarily defined by a single loop that is significantly larger in ImpA than it is in either SmE or AM0627. In our docked structure, we observed that the short loop of SmE allowed the enzyme to easily accommodate the P1 glycan, with neither the GalNAc nor the Gal residue forming direct contacts with the enzyme; the sialic acid residue could interact with the enzyme or project toward solvent[18,47,50]. The long loop in ImpA, by contrast, sterically clashes with all three subunits of the P1 glycan—explaining why ImpA is unable to cleave between adjacent residues bearing glycosylation. More broadly, these and prior findings suggest a delicate interplay between the hairpin and loop in determining this enzyme family's tolerance, preference, or requirement for particular glycans at P1 (Supplementary Fig. 13). In the case of SmE, the short loop and absence of a hairpin allows the enzyme to tolerate (but not require) larger glycans at the P1 position, which again supports our MS findings. For a discussion regarding accessory mucin binding domains in various O-glycoproteases, please see the Supplemental Information (Supplementary Figs. 14, 15).

## Glycoproteomic mapping of TIM-1, −3, and −4

With this tool in-hand, we reasoned that SmE could be used to sequence immune regulatory mucin-domain glycoproteins at the molecular level. In particular, TIM-1, -3, and -4 are key players in immune cell regulation and are predicted to be modified at many O-glycosylation sites[53]. Each protein contains an N-terminal variable immunoglobulin (IgV) domain followed by a densely glycosylated mucin domain of varying length, a single transmembrane domain, and a C-terminal intracellular tail[54]. TIM-3 is highly implicated in the regulation of anti-tumor immunity and is being developed as a target in cancer immunotherapy, thus much of the literature to date has focused on better understanding its molecular interactions. In short, when TIM-3 is not bound to its extracellular ligands via its IgV domain, the TIM-3 cytoplasmic tail is thought to recruit cytoplasmic kinases that promote activation of the T cell receptor (TCR), in turn increasing T cell proliferation and survival. When TIM-3 is engaged by its ligands, however, phosphorylation of the TIM-3 cytoplasmic tail reverses this effect to promote a state of T cell exhaustion characteristic of the immune microenvironment of many cancers (Fig. 3A, left)[55]. As such, several antibodies against TIM-3 are currently being investigated as cancer immunotherapies, often in combination with canonical checkpoint inhibitors like PD-1[56–58].

Compared to TIM-3, less is known about TIM-1 and TIM-4, potentially because these proteins are predicted to bear more O-glycosites than TIM-3, which complicates their analysis. However, it is known that the combination of TIM-1 blockage and TCR stimulation promotes T cell proliferation and cytokine production;[59] TIM-4 is a PtdSer receptor and binds PtdSer exposed on the surface of apoptotic cells (Fig. 3A, right)[60]. While much remains to be discovered about TIM-1 and -4, it is apparent that the entire TIM family plays critical roles in regulating immune responses in normal and dysregulated states. However, only predicted glycosylation sites in the TIM family have been discussed in the literature, leaving their true glycoproteomic landscape a mystery. It follows, then, that we also do not understand how glycosylation contributes to TIM protein–ligand binding, structural dynamics, and intracellular signaling. Ultimately, this lack of information hampers understanding of TIM family structure and function, which could have strong implications for cancer immunotherapy.

According to NetOGlyc 4.0[61], TIM-1, -3, and -4 were predicted to bear 67, 8, and 66 O-glycosites, respectively. Additionally, we recently developed a "Mucin Domain Candidacy Algorithm" that takes into account predicted O-glycosites, glycan density, and subcellular location in order to output a "Mucin Score"[7]. This value was developed as a method to gauge the likelihood that a protein contains a mucin domain; a Mucin Score above 2 indicated a high probability. Interestingly, TIM-1 and TIM-4 scored above 6, whereas TIM-3 received a score of 0[7]. Beyond the biological implications of these proteins, we were curious to understand the glycoproteomic landscapes of the TIM family given the large disparity between proteins in predicted O-glycosites and Mucin Scores. We therefore digested the three recombinant TIM protein ectodomains with SmE and performed MS analysis followed by manual curation of the glycopeptides. Given our earlier observations regarding SmE's resistance to proteins bearing sparse glycosylation, we digested TIM-3 with ImpA and OgpA to ensure full sequence coverage and O-glycosite identification. As seen in Fig. 3B, with a full list of annotated glycopeptides in Supplementary Data 4, 7-8, we identified all 67 of the predicted O-glycosites on TIM-1; many of these sites were modified by a myriad of O-glycans, thus demonstrating the massive microheterogeneity in mucin domains.

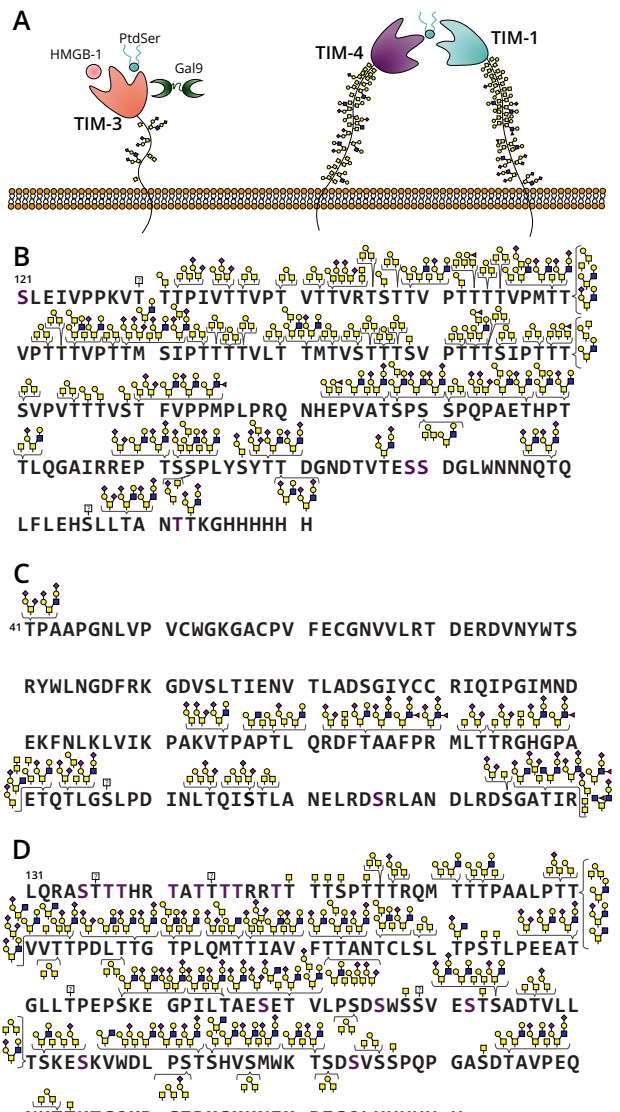

**Fig. 3 | Glycoproteomic mapping of TIM family proteins. A** Cartoon of TIM family structure and ligand interactions. TIM-3 interacts with its ligands PtdSer, HMGB1, CEACAM1, and/or Gal-9; through intracellular signaling these interactions deactivate T cell function and cytokine release. TIM-1 and TIM-4 purportedly interact through PtdSer to enact effector function. Recombinant TIM-1 (**B**), TIM-3 (**C**), and TIM-4 (**D**) were subjected to digestion with SmE, ImpA, OgpA, and/or trypsin followed by MS analysis and manual data interpretation. Brackets indicate glycans sequenced at each Ser/Thr residue at >5% relative abundance. For full glycoproteomic sequencing data, see Supplementary Data 4, 7, and 8.

TIM-4 was similarly dense in glycosylation, and we site-localized a total of 51 O-glycosites (Fig. 3D). In contrast, only 14 sites of O-glycosylation were detected on TIM-3; the glycosylation was also much less dense, although still quite heterogeneous (Fig. 3C). Generally, recombinantly expressed proteins are thought to display relatively simple glycosylation (e.g., core 1 or 2 structures). Intriguingly, despite the fact that these proteins were recombinantly expressed in HEK293 cells, we observed not only these glycans, but also highly sialylated and fucosylated structures.

## MD simulations of TIM-3 and -4 elucidate the structural and dynamical impacts of glycosylation

Following the initial glycoproteomic mapping, we asked how these different glycosylation patterns could affect the structures, and

potentially functions, of the TIM proteins. Although the IgV domain structures have been solved via X-ray crystallography and NMR, the mucin domains were excluded from this analysis, presumably due to the high heterogeneity and density of O-glycosylation[62,63]. After failed attempts to perform cryoEM on the full TIM-3 and -4 ectodomains, we reasoned that molecular modeling and MD simulations could be an alternative method to predict mucin domain structure and to better understand how glycosylation contributes to the dynamic properties of these proteins. As described above, the microheterogeneity of each glycosite was incredibly high; thus, in order to accurately reconstruct the TIM glycoproteomic landscape, we needed to identify the most abundant glycan at each residue. To do so, we generated extracted ion chromatograms (XICs) of every glycopeptide detected from TIM-1, -3, and -4 to calculate area-under-the-curve relative quantitation using Thermo Xcalibur (Supplementary Data 4, 7–8). The relative abundance of every detected glycan, at each O-glycosite, is depicted in Fig. 4A (right). As others before have suggested, the glycan size and heterogeneity were much lower in areas of dense glycosylation; sparse O-glycosites afforded larger and more diverse glycan structures[64,65].

By obtaining the most abundant O-glycan at each glycosite, we built two all-atom computational models of the fully glycosylated transmembrane glycoproteins TIM-3 and TIM-4 (Fig. 4A, left) to better understand the contribution of glycan density on the overall flexibility and length of TIMs. Each of these systems contained their respective globular IgV domain, mucin-domain, α-helical transmembrane domain, and cytoplasmic tail. Approximately 700 and 830 ns of simulation data were generated for TIM-3 and -4, respectively (see SI Methods for full simulation details).

To identify the degree to which TIM-3 and TIM-4 mucin domains compress during simulation, we calculated a normalized end-to-end distance for each protein's mucin domain as a function of time (Fig. 4B). The results indicate that the TIM-3 mucin domain, with only 14 glycans and 71 amino acids, compresses far more significantly than the TIM-4 mucin domain, which contains 51 glycans and 179 amino acids. We quantified this change by calculating persistence length, defined as the distance (in Å) at which the motions of two monomers along a polymeric chain become decorrelated from one another. Strong intramolecular interactions within monomers can lead to highly correlated motions along the polymeric chain, overcoming energetic gains of interactions with solvent or enhanced conformational degrees of freedom, and thus long persistence lengths. Using data from our MD simulations, we calculated the persistence lengths of the TIM-3 and -4 mucin domains to be $81 \pm 24$ Å and $415 \pm 10$ Å, respectively. Thus, these two mucin domains have drastically different degrees of correlation within their protein backbones, likely originating from their varied degrees of glycosylation (Fig. 4B, inset).

During initial analysis and trajectory visualization, we noticed that the TIM-3 mucin domain underwent a significant degree of bending such that the IgV domain tilted toward the membrane. To better quantify this, we calculated the angle between two vectors for both TIM-3 and -4 mucin domains: one drawn from the central residue to the first residue, and another drawn from the central residue down to the most C-terminal residue (Fig. 4D, see SI Methods for complete details). We observed that the TIM-4 mucin domain largely sampled bending angles close to 180°, i.e., the TIM-4 mucin domain was largely linear and "bottle-brush like". The TIM-3 mucin domain, by contrast, bent quite significantly and sampled a large range of different angles with similar probabilities (Fig. 4E, F). It is important to note that, although the results extracted from the simulations suggest differences between the two proteins, our observations concerning the persistence length and bending angles are not fully converged due to limitations on time and computing power, so should be considered preliminary in nature.

To investigate the effect of variable glycosylation on mucin identity and functional dynamics, we aimed to quantify total versus

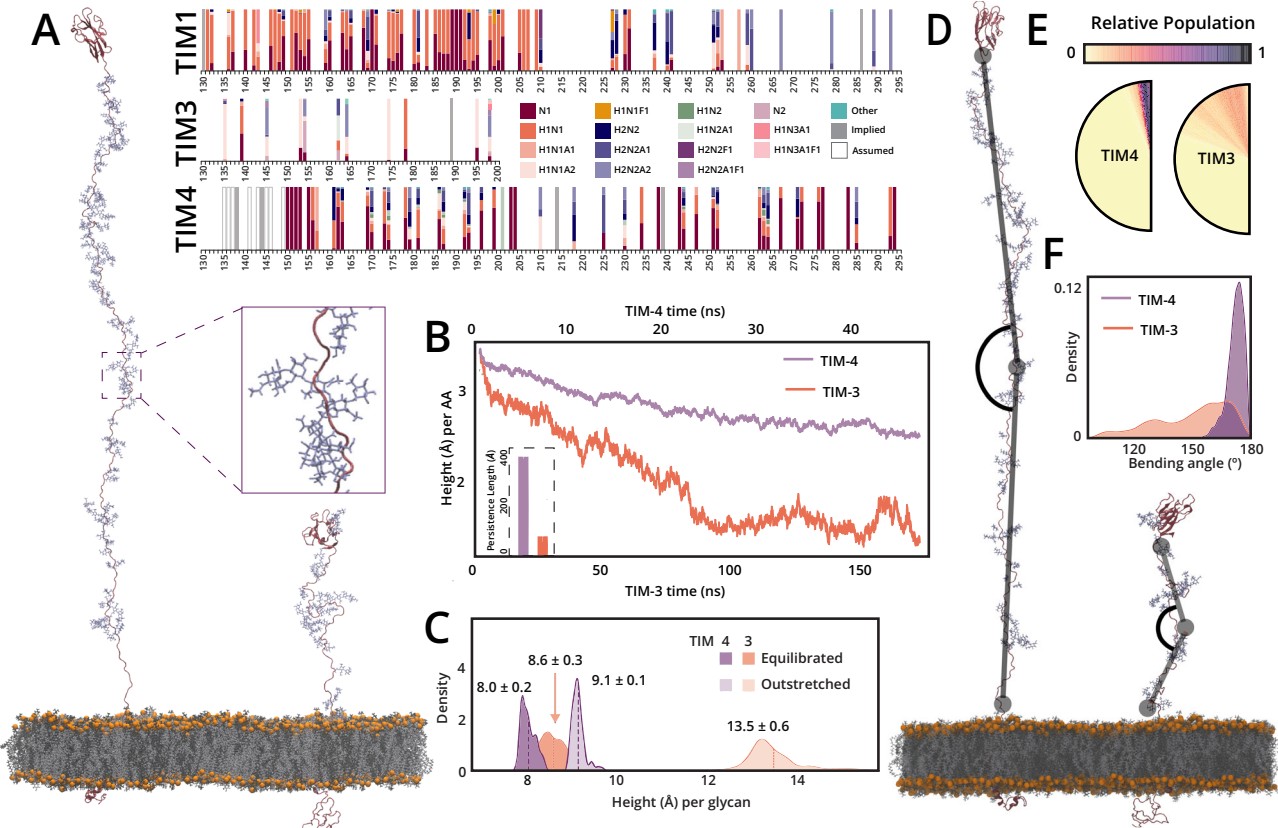

**Fig. 4 | MD simulations of TIM-3 and -4 elucidate the structural and dynamical impact of glycosylation. A** (right) XICs were generated for each glycopeptide from TIM-1, -3, and -4 and area-under-the-curve quantitation was performed; glycan composition color legend shown in center. N - HexNAc (GalNAc), H - hexose (galactose), A - NeuAc (sialic acid), F - fucose. (A, left) Image detailing TIM-4 and TIM-3 models along with an inset view highlighting the dense TIM-4 glycosylation. **B** End-to-end distance of TIM-3 and TIM-4 mucin domains normalized by total length (number of amino acids, AA) within the mucin domains, plotted as a function of simulation length. (inset) Persistence length calculated for TIM-3 and TIM-4 from all simulation replicas. **C** Histograms detailing the end-to-end distance of TIM-3 and TIM-4 mucin domains, normalized by total number of glycans, in the outstretched (starting) conformation (lighter distributions) and equilibrated conformation (darker distributions). **D** Image demonstrating the "bending angle" as calculated in the following panels. **E** Semi-circles graphically detailing the bending angles visited over the complete course of simulations for TIM-3 and -4, angles colored according to relative population. **F** Histograms detailing bending angles sampled by TIM-3 and -4 over the course of all simulations. Source data are provided in the Source Data file.

effective glycosylation in a thoroughly glycosylated mucin domain (as in TIM-4) versus in a sparsely glycosylated mucin domain (as in TIM-3). Herein, we define the total glycosylation as the ratio of the length of the outstretched, unequilibrated mucin domain protein backbone to the total number of glycans. Similarly, we define effective glycosylation as the ratio of the length of a relaxed, equilibrated mucin domain protein backbone to the total number of glycans. These two values thus illustrate the "height per glycan (Å)" under outstretched and relaxed conditions. As shown in Fig. 4C, the height per glycan in heavily glycosylated TIM-4 remains nearly the same in both outstretched and equilibrated states: $9.1 \pm 0.1$ Å and $8.0 \pm 0.2$ Å, respectively. This indicates that upon relaxation of the mucin domain protein backbone, O-glycans still maintain a similar distribution relative to one another as in the fully outstretched case, i.e., total glycosylation equals effective glycosylation. However, for TIM-3, the height per glycan distance drops significantly following equilibration, going from $13.5 \pm 0.6$ Å to $8.6 \pm 0.3$ Å. In fact, following equilibration, this height per glycan distance seen in TIM-3 becomes similar to those distributions seen in TIM-4. Through trajectory visualization, specific glycan-glycan pairs in TIM-3 were found to be responsible for a large portion of the decrease in height per glycan distance. Specific distant pairs (≥3 glycans away from one another) of glycans are seen to interact via hydrogen bonding, almost as if these glycans are "holding hands," as exemplified by the TIM-3 glycan pair G7 and G4 (glycosylation sites T145 and T162, respectively; Supplementary Fig. 16). These results demonstrate the

power of MD simulations in characterizing members of the mucinome, as glycoproteomic mapping alone cannot provide atomic-level structural insight, including conformational changes that may allow distant O-glycan pairs to find each other and reach new, functionally significant conformations.

## Functional consequences of TIM protein mucin domains

Beyond the differences in structural rigidity and extension imparted by altered density of O-GalNAc glycosylation[66,67], we sought to uncover other ways in which variable glycosylation of mucin domains might influence protein function. Indeed, our MD simulations revealed that TIM-4 extended approximately 5-fold further from the cell surface than TIM-3 despite having only 50% more amino acids in the extracellular region. Differences in the protrusion of these receptors from the cell surface should impact how accessible they are to their ligands. The immune cell glycocalyx is dominated by CD43 and CD45, which extend up to 45 and 51 nm from the cell surface, respectively[68-70]. With a persistence length of ~41.5 nm, TIM-4 could span much of this distance, bringing its IgV domain close to the external environment. Conversely, the less glycosylated TIM-3 extracellular region relaxes to only ~8.1 nm, suggesting that it would remain obscured by the surrounding glycocalyx. As a result, PtdSer, a ligand for both TIM-3 and TIM-4, might more easily engage TIM-4 on immune cells to alter cell signaling. Interestingly, TIM-3 has multiple ligands, and it has been proposed that these ligands can all interact with TIM-3 simultaneously,

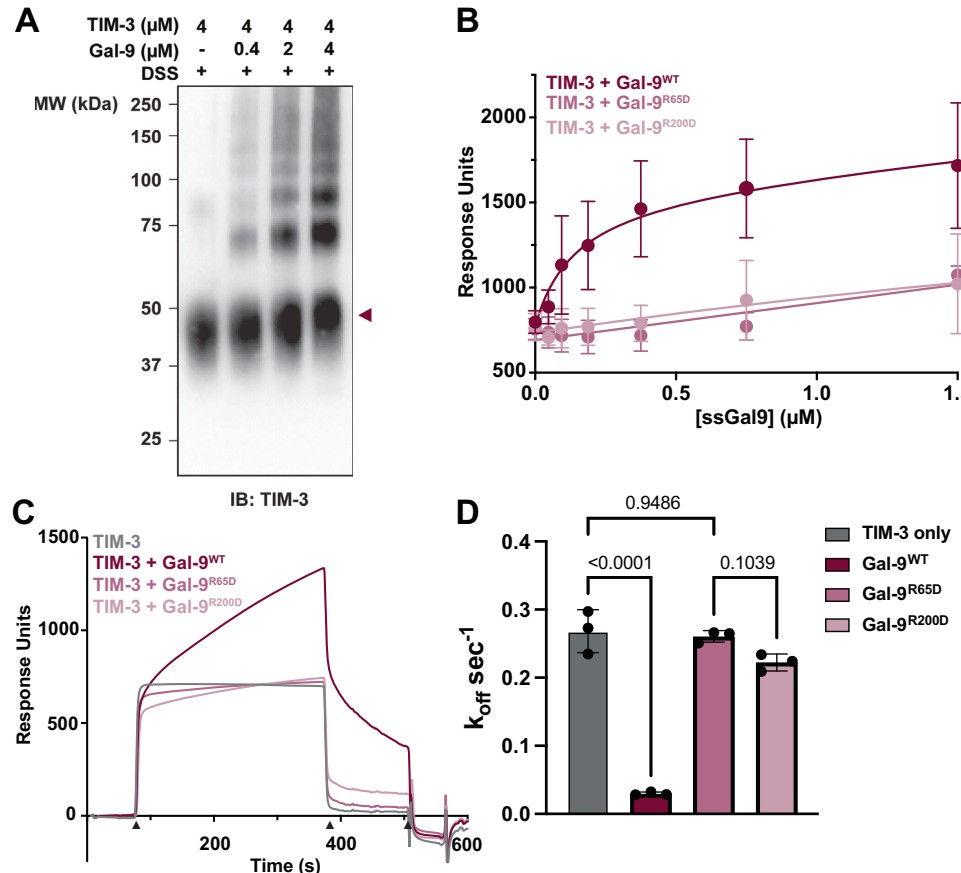

**Fig. 5 | Gal-9 crosslinks TIM-3 to enhance affinity to PtdSer. A** Gal-9 and the extracellular region of TIM-3 were incubated in the presence of the crosslinking reagent disuccinimidyl suberate (DSS). Western blotting for TIM-3 indicated that higher molecular weight species formed with increasing concentration of Gal-9. **B** Surface plasmon resonance was used to quantify the impact of (bivalent) Gal-9[WT], with those of monovalent Gal-9[R200D], or Gal-9[R65D] variants on the binding of TIM-3 to immobilized membranes containing 20% PtdSer and 80% phosphatidylcholine. Lipid vesicles were immobilized on an L1 sensorchip, and protein samples were flown over the surface with 1 mM CaCl₂. The increase in response units (RU) observed for TIM-3 in the presence of Gal-9[WT], Gal-9[R65D], or Gal-9[R200D] is plotted as a function of Gal-9 concentration. Data are presented as mean +/- SD for n = 3 independent experiments. (**C**) Sensorgrams of TIM-3, TIM-3 and Gal-9[WT], TIM-3 and Gal-9[R65D], and TIM-3 and Gal-9[R200D]. The black triangles below the x-axis indicate the start of the sample injection (~75 s), the end of the sample injection (~375 s), and the regeneration (~510 s). Data points from the dissociation phase, (~375–500 s), were fit using a nonlinear regression model that was used to estimate $k_{off}$ values. (**D**) Bar graph showing the calculated $k_{off}$ values as mean ± SD for $n = 3$ independent experiments at the various conditions tested. Significance was tested using one-way ANOVA in Graphpad PRISM. Source data are provided as a Source Data file.

potentially altering the binding interactions seen with individual isolated ligands[11,54]. For instance, the TIM-3 ligand Gal-9 is a bivalent lectin that binds glycoproteins decorated with the LacNAc (Gal-GlcNAc) structure and has been shown to crosslink receptors at the cell surface[71,72]. We hypothesized that by employing both of its carbohydrate recognition domains (CRDs), Gal-9 can create networks of TIM-3 at the cell surface. Further, these lattices could result in patches of more accessible TIM-3, thereby increasing affinity (and avidity) for other ligands like PtdSer.

To test this hypothesis, we assessed the ability of Gal-9 to crosslink TIM-3 by incubating the bivalent lectin with the complete extracellular region of TIM-3 in the presence of the chemical crosslinking reagent disuccinimidyl suberate (DSS). For these experiments, we employed the variant of Gal-9 developed by Itoh and colleagues to enhance stability and solubility of the recombinantly expressed protein[73]. Only when TIM-3 and Gal-9 were incubated in the presence of DSS were larger molecular weight complexes observed above the band for monomeric TIM-3 (Fig. 5A, lanes 2-4, Supplementary Fig. 18). As the concentration of Gal-9 increased the bands of these higher molecular weight species intensified, indicating that the shift of TIM-3 into complexes was dependent on the amount of Gal-9 present. Furthermore, the laddered pattern of

TIM-3 suggests the formation of a lattice that includes these two binding partners.

We next asked whether oligomerization of TIM-3 by Gal-9 might enhance its avidity for PtdSer in cell membranes. Our previous findings demonstrated that TIM-3 binds PtdSer-containing membranes, and that mutations that impair TIM-3 binding to PtdSer reduce its impact on T cell signaling[74]. Using surface plasmon resonance (SPR), we quantified the TIM-3 binding to PtdSer-containing membranes in the presence and absence of wildtype Gal-9 (Gal-9[WT]) or variants of Gal-9 with mutations in one CRD (Gal-9[R65D] or Gal-9[R200D]) that render the protein monovalent—which should reduce crosslinking of TIM-3. Adding Gal-9[WT] to TIM-3 enhanced its binding to PtdSer-containing surfaces in a dose-dependent manner (Fig. 5B), whereas the mutated Gal-9 variants did not, indicating that the increased PtdSer binding affinity reflects the formation of a TIM-3/Gal-9 lattice.

We were not able to quantify the effects of Gal-9-induced crosslinking on the affinity of TIM-3 for PtdSer because of problems with Gal-9 precipitation at high concentrations. We therefore exploited the SPR approach to perform kinetic analysis and calculate $k_{off}$ (the rate of dissociation) from PtdSer-containing membranes for the different TIM-3/Gal-9 complexes. Following rapid association with the surface, equilibrium binding was reached for all samples except TIM-3 with Gal-9[WT]

within 100–350 s (Fig. 5C). The continued escalation of signal for the TIM-3/Gal-9$^{WT}$ sample likely reflects continued growth of a TIM-3/Gal-9 lattice on the PtdSer-containing membranes immobilized on the sensorchip. Moreover, Gal-9$^{WT}$ significantly slowed the dissociation of TIM-3 from the PtdSer-containing surface when compared with TIM-3 alone or TIM-3 bound to monovalent Gal-9 variants (Fig. 5C). The resulting curves were fit with a one-phase exponential decay model to estimate $k_{off}$ and quantify the effect of Gal-9$^{WT}$ on TIM-3/PtdSer binding (Fig. 5D). We found that $k_{off}$ for TIM-3 in the presence of Gal-9$^{WT}$ ($0.030 \pm 0.002$ s$^{-1}$) was ~9-fold slower than $k_{off}$ for TIM-3 alone ($0.27 \pm 0.03$ s$^{-1}$). Mutation in either of the two CRDs in Gal-9 abrogated this effect, with $k_{off}$ for TIM-3 with the mutant Gal-9 variants being unchanged ($0.26 \pm 0.01$ s$^{-1}$ for Gal-9$^{R65D}$ and $0.22 \pm 0.01$ s$^{-1}$ for Gal-9$^{R200D}$). These data argue that Gal-9$^{WT}$ significantly enhances the avidity of TIM-3 for PtdSer through crosslinking multiple TIM-3 molecules and enhancing their access to PtdSer on an opposing membrane.

Although Gal-9 is known to interact with other immune modulators[75–79], no experimental evidence suggests Gal-9 binds to TIM-4. TIM-4 has a substantially higher binding affinity for PtdSer than that of TIM-3[74], and also extends further from the cell surface than TIM-3 as a result of its increased density of glycosylation, according to our analysis. Thus, TIM-4 likely does not require lattice formation to increase local receptor concentration and improve binding strength. While these findings are preliminary in nature, they shed new potential light on how mucin-type glycosylation impacts the structure and function of important cellular receptors. In the case of TIM-3, we suggest that the relatively sparse glycosylation allows bivalent lectin binding to control the avidity of an otherwise low-affinity PtdSer binder for immune cell control. For TIM-4, which has a high-affinity PtdSer-binding site, dense glycosylation can project it from the glycocalyx in a way that allows it to constitutively bind PtdSer, reflecting the differences in TIM-3 and TIM-4 function.

## Discussion

Historically, numerous challenges have impeded the study of mucin-domain glycoproteins; however, new tools continue to be introduced to unveil mucin glycosylation status, functional roles, and biological impact. Here, we present an addition to this toolkit and use it to better understand mucin-domain glycoprotein structure and dynamics. We first thoroughly characterized the mucinase SmE, which demonstrated a uniquely broad cleavage motif and outperformed commercially available O-glycoproteases, thus enabling unprecedented glycoproteomic mapping of biologically relevant mucin proteins. In particular, we elucidated the glycosylation landscape of clinically relevant immune checkpoint proteins TIM-1, -3, and -4 and used this information to enable glycoproteomic-guided MD simulations. The data afforded by SmE treatment, in concert with MD simulations, has opened the door to a realm of atomic-level insight into mucin-domain-containing proteins, their structure-function relationships, roles of glycosylation, and the recognition mechanisms of bacterial mucinases. Ultimately, we developed a powerful workflow to understand detailed molecular structure and guide functional assays for all members of the mucinome.

That said, we have only begun to unlock the potential of this workflow, especially as it pertains to the TIM family of proteins. To be sure, aberrant glycosylation is a hallmark of cancer and typical O-glycosylation changes involve truncation of normally elaborated glycan structures[80]. Such shortened glycans could strongly impact the "linearity" of the mucin backbone, thus changing TIM protein protrusion from the glycocalyx and ultimately modulating function. As such, transformed O-glycosylation could have implications for how the glycoproteins interact with each other, their ligands, and as a result, intracellular signaling and T cell cytotoxicity. Future efforts will be devoted to glycoproteomic mapping of endogenous TIM proteins from primary T cells and patient samples to discover how glycan

structures change in health and disease. In concert with biological assays, we will use this information to drive MD simulations that probe how altered glycosylation could affect ligand binding, intracellular interactions, and antibody recognition. Beyond the TIM family of glycoproteins, many other glyco-immune checkpoints have emerged as prominent mechanisms of immune evasion and therapeutic resistance in cancer;[31] we envision that our workflow will also help elucidate structure-function relationships in these proteins.

Aside from MS analysis, SmE has the potential to make a larger impact on the field of glycobiology. In our previous work, we used StcE for clearance of mucins from the cell surface and determined that Siglec-7, but not Siglec-9, selectively bound to mucin-associated sialoglycans[17]. We then upcycled an inactive point mutant of StcE to develop staining reagents for Western blot, immunohistochemistry, and flow cytometry[18]. We also took advantage of the mutant StcE to develop an enrichment procedure that allowed for the selective pull-down of mucin glycoproteins[7]. Finally, and most recently, an engineered version of StcE conjugated to nanobodies was used for targeted degradation of cancer-associated mucins[81]. Given that SmE is similarly active on live cells, has a complementary cleavage motif, two mucin binding domains, and potentially different endogenous targets, future work will be aimed at investigating whether SmE can augment our current mucinase toolkit and therapeutic strategies.

Previously, we developed an algorithm to help identify proteins that have a high probability of bearing a mucin domain[7]. While we recognized that our definition of a mucin domain was novel but rudimentary, the present work has confirmed that our understanding of mucin domains is incomplete. TIM-1 and -4 were predicted by our algorithm to be high-confidence mucins, whereas TIM-3 received a score of 0. Here, our glycoproteomic mapping combined with MD simulations demonstrated that absolute glycosylation (i.e., O-glycans per amino acid residue) can be dramatically different than effective glycosylation (i.e., O-glycosylation in relation to total surface area after folding). Thus, while density of O-glycosylation can absolutely be an indication that a mucin domain is present, it does not reveal the entire story, and our definition of a mucin domain continues to develop— possibly requiring subdivision according to the role that high-density versus low-density glycosylation plays. While it would be ideal to obtain high-resolution structures of these mucin-domain glycoproteins, that is likely a long-term objective that will challenge structural biology methods. In the meantime, our workflow is a tangible mechanism for visualizing the enigmatic mucin family to not only better understand the definition of a mucin domain, but to also study the structural dynamics and functions that lie within.

## Materials and methods
### Materials
Recombinantly expressed TIM-1 (AAC39862) and TIM-4 (Q96H15) for glycoproteomic studies were purchased from R&D Systems (9319-TM, 9407-TM), whereas recombinant extracellular protein for functional studies was made in house (see below). For structural characterization, TIM-1 was purchased from R&D systems (11157-TM) and TIM-4 was purchased from LifeSpan Biosciences (LS-G139224). TIM-3 (Q8TDQ0) was purchased from LifeSpan Biosciences (LS-G97947). CD43 (P16150) and TIM-1 recombinantly expressed in NS0 cells were purchased from R&D systems (9680-CD,1750-TM). C1-Inh (P05155) and fibronectin (P02751) isolated from human plasma were purchased from Sigma Aldrich (E0518, F1056). Bovine Fetuin-A (P12763) was purchased from Promega (V4961). GP1bα (P07359) was isolated as described previously[8]. The plasmids for His-tagged pET28a-SmEnhancin and recombinant StcE protein were kindly provided by the Bertozzi laboratory.

### Expression and purification of SmEnhancin
For pET28a-SmEnhancin plasmid extraction, a 50 μL aliquot of chemically competent *E. coli* DH5α cells (NEB, C2988J) was thawed on ice.

Approximately 100 ng of plasmid was added to the cells and incubated on ice for 30 min. Cells were then transformed by heat-shock at 42 °C for 30 s. Room temperature SOC media (Invitrogen, 15544-034) was added (950 μL) to the cells then incubated at 37 °C with agitation at 250 rpm for 1 h. A 150 μL aliquot of the transformed cells was then transferred to a LB-agar (Fisher, BP1425) plate with kanamycin (Sigma Aldrich, K1377) and incubated overnight at 37 °C. Kanamycin was used throughout at a final concentration of 50 μg/mL. A single colony was picked and used to inoculate an overnight culture of 100 mL Luria broth (LB) (Sigma Aldrich, L3022) with kanamycin. The culture was incubated at 37 °C with agitation at 250 rpm. pET28a-SmEnhancin plasmid DNA was extracted using a Qiagen Plasmid Midi Kit (Qiagen, 12143) using the protocol provided by the manufacturer. DNA concentration was determined by NanoDrop One Microvolume UV-Vis Spectrophotometer (Thermo-Fisher) then stored at −80 °C. After extraction, the DNA sequence was verified using Plasmidsaurus.

For protein expression of SmEnhancin (SmE), a 20 μL aliquot of competent *E. coli* BL21(DE3) cells (Millipore Sigma, 69450-4) was thawed on ice. Approximately 10 ng of pET28a-SmEnhancin was added to cells and incubated on ice for 5 min. Cells were then transformed by heat-shock at 42 °C for 30 s. Room temperature SOC media was added (80 μL) to the cells then incubated at 37 °C with agitation at 250 rpm for 1 h. A 100 μL aliquot of the transformed cells was then transferred to a LB-agar plate with kanamycin and incubated overnight at 37 °C for colony growth. A single colony was picked to inoculate a 10 mL overnight culture of LB with kanamycin and incubated at 37 °C with agitation at 250 rpm. A glycerol stock was made by mixing 4 mL of the overnight culture with 4 mL of 50% glycerol (Sigma Aldrich, G7893) and stored at −80 °C. The remaining overnight culture was used to inoculate a 1 L LB culture with kanamycin. This culture was also incubated at 37 °C with agitation at 250 rpm until it reached an optical density of 0.6-0.8. The maxi culture was then induced with a final concentration of 0.1 mM isopropyl-β-D-1-thiogalactopyranoside (IPTG) (American Bio, AB00841) and grown overnight at 16 °C with agitation at 250 rpm. The bacterial cells were harvested by centrifugation at 3000*g* for 45 min at 4 °C. The supernatant was decanted and the cell pellet was stored at −80 °C until lysis was performed.

The cell pellet was lysed in a buffer containing 20 mM Tris-HCl at pH 8 (Thermo Scientific, J3636.K2), 200 mM NaCl (Fisher Scientific, S25877), 2 mM magnesium chloride ($MgCl_2$) (American Bio, AB09006), 10% glycerol, and 125 U/mL Benzonase nuclease (Sigma Aldrich E1014). Additionally, 100 μg/mL lysozyme (Thermo Scientific, 89833) and 1% Triton X-100 (Alfa Aesar, J66624) were added to aid cell lysis. For inhibition of protease activity, a cOmplete Mini EDTA-free protease inhibitor cocktail (Roche, 11836170001) was used alongside 1 mM phenylmethylsulfonyl fluoride (PMSF) (American Bio, AB01620). Cells were then resuspended using 1 mL of chilled lysis buffer per gram of cells. The cell suspension was further homogenized by five pulses of probe sonication with 5 s of sonication at 35% amplitude followed by 15 s pauses (QSonica Q500). The solution was kept on ice throughout sonication to prevent protein denaturation/degradation. Lysate was clarified by spinning at 25,000*g* for 45 min at 4 °C and the supernatant was filtered sequentially through 0.45 μm (Millipore, SLHAR33SS), and 0.2 μm (Cytiva, 10462300) syringe filters.

The protein was purified using an ÄKTA Pure FPLC (Cytiva) with a HisTrap HP column (Cytiva, 17524801). The column was equilibrated for 5 column volumes (CV) at 1 mL/min using buffer A (20 mM Tris-HCl pH8, 500 mM NaCl, 25 mM imidazole (Sigma Aldrich, I202)) prior to loading the sample at a flow rate of 0.5 mL/min. A conditional wash was then performed, rinsing at 1 mL/min with buffer A until the absorbance of the column flowthrough fell below 10 mAU. The protein was then eluted with a 15 CV linear gradient to 100% buffer B (20 mM Tris-HCl pH8, 500 mM NaCl, 500 mM imidazole). During sample load and wash phases, 5 mL fractions of the eluent were collected, while 2 mL

fractions were collected during the elution. Fractions containing pure protein were identified by SDS-PAGE gel (BioRad, 3450123). Amicon Ultra 30 kDa MWCO filters (Millipore Sigma, UFC803024) were then used to combine, concentrate, and buffer exchange the purified protein fractions into 10 mM Tris, pH 7.4 (American Bio, AB14044). Protein concentration was determined by NanoDrop One Microvolume UV-Vis Spectrophotometer (Thermo-Fisher) before storage at −80 °C.

### Mucinase digestion
All proteins were first digested with either SmE, ImpA (NEB, P0761), or OgpA (Genovis, G2-OP1-020), prior to any further processing. All solutions were prepared using MS-grade water (Thermo Scientific, 51140). For the structural characterization and mapping of TIM-1 and TIM-4 as well as the analysis of GP1bα, 6–8 μg of protein were used for each digest. All other digests were performed using 2 μg of protein. Each glycoprotein was digested with the O-glycoproteases in a total volume of approximately 15 μL of fresh 50 mM ammonium bicarbonate (AmBic) (Honeywell Fluka, 40867) overnight at 37 °C. Digestions with SmE were conducted at an enzyme-to-substrate ratio of 1:10, while ImpA and OgpA were digested according to commercial instructions. When sialidase was used, it was added alongside the O-glycoproteases at concentrations in accordance with the manufacturer's protocol (NEB, P0720).

### SDS-PAGE analysis
Fractions from SmE protein purification were run on a 4–12% Criterion XT BisTris gel (Bio-Rad, 3450123) in MES XT buffer (Bio-Rad, 1610789) at 180 V for 60 min, alongside Precision Plus All Blue Protein Standard (Bio-Rad, 1610373). Digested proteins were separated on a 4–12% Criterion XT BisTris gel (Bio-Rad, 3450123) in MOPS XT buffer (Bio-Rad, 1610788) at 180 V for 60 min, alongside Blue Easy Protein Ladder (NIPPON Genetics, MWP06). All protein gels were imaged on a LiCOR Odyssey instrument following a 30 min incubation with Aquastain (Bulldog Bio, AS001000).

### Mass spectrometry sample preparation
After mucinase digestion, dithiothreitol (DTT) (Sigma Aldrich, D0632) was added to a concentration of 2 mM and allowed to react at 65 °C for 20 min followed by alkylation in 5 mM iodoacetamide (IAA) (Sigma Aldrich, I1149) for 15 min in the dark at room temperature.

For samples where the presence of N-glycosylation interfered with the identification of potential O-glycosylation sites, a PNGaseF (NEB, P0705) digestion was performed. This included the characterization of C1-Inh, while secondary files for TIM-1 and TIM-3 were generated after gaps in coverage indicated N-glycosylation. The concentrated enzyme was diluted 1:10 and 1 μL of the diluted stock was used for each 2 μg of protein. After allowing the enzyme to react overnight, the protein was buffer exchanged into 50 mM AmBic using 10 kDa MWCO filters (Merck Millipore, UFC501024).

Proteins with fewer sites of glycosylation (i.e. Fetuin, TIM-3, and C1-Inh) underwent an additional digestion by adding sequencing-grade trypsin (Promega, V5111) in a 1:50 enzyme:substrate (E:S) ratio for 6 h at 37 °C. All reactions were quenched by adding 1 μL of formic acid (Thermo Scientific, 85178) and diluted to a volume of 200 μL prior to desalting. Desalting was performed using 10 mg Strata-X 33 μm polymeric reversed phase SPE columns (Phenomenex, 8B-S100-AAK). Each column was activated using 500 μL acetonitrile (ACN) (Honeywell, LC015) followed by 500 μL 0.1% formic acid, 500 μL 0.1% formic acid in 40% ACN, and equilibration with two additions of 500 μL 0.1% formic acid. After equilibration, the samples were added to the column and rinsed twice with 200 μL 0.1% formic acid. The columns were transferred to a 1.5 mL tube for elution by two additions of 150 μL 0.1% formic acid in 40% ACN. The eluent was then dried using a vacuum concentrator (LabConco) prior to reconstitution in 10 μL of 0.1% formic acid.

## Mass spectrometry data acquisition

Samples were analyzed by online nanoflow liquid chromatography-tandem mass spectrometry using an Orbitrap Eclipse Tribrid mass spectrometer (Thermo Fisher Scientific) coupled to a Dionex UltiMate 3000 HPLC (Thermo Fisher Scientific). For each analysis, 4 µL was injected onto an Acclaim PepMap 100 column packed with 2 cm of 5 µm C18 material (Thermo Fisher, 164564) using 0.1% formic acid in water (solvent A). Peptides were then separated on a 15 cm PepMap RSLC EASY-Spray C18 column packed with 2 µm C18 material (Thermo Fisher, ES904) using a gradient from 0-35% solvent B (0.1% formic acid with 80% acetonitrile) in 60 min.

Full scan MS1 spectra were collected at a resolution of 60,000, an automatic gain control (AGC) target of 3e5, and a mass range from 300 to 1500 $m/z$. Dynamic exclusion was enabled with a repeat count of 2, repeat duration of 7 s, and exclusion duration of 7 s. Only charge states 2 to 6 were selected for fragmentation. MS2s were generated at top speed for 3 s. Higher-energy collisional dissociation (HCD) was performed on all selected precursor masses with the following parameters: isolation window of 2 $m/z$, 29% normalized collision energy, orbitrap detection (resolution of 7,500), maximum inject time of 50 ms, and a standard AGC target. An additional electron transfer dissociation (ETD) fragmentation of the same precursor was triggered if 1) the precursor mass was between 300 and 1,500 m/z and 2) 3 of 8 HexNAc or NeuAc fingerprint ions (126.055, 138.055, 144.07, 168.065, 186.076, 204.086, 274.092, and 292.103) were present at ± 0.1 $m/z$ and greater than 5% relative intensity. Two files were collected for each sample: the first collected an ETD scan with supplemental energy (EThcD) while the second method collected a scan without supplemental energy. Both used charge-calibrated ETD reaction times, 100 ms maximum injection time, and standard injection targets. EThcD parameters were as follows: Orbitrap detection (resolution 7500), calibrated charge-dependent ETD times, 15% nCE for HCD, maximum inject time of 150 ms, and a standard precursor injection target. For the second file, dependent scans were only triggered for precursors below 1000 $m/z$, and data were collected in the ion trap using a normal scan rate.

## Mass spectrometry data analysis

Raw files were searched using O-Pair search with MetaMorpheus against directed databases containing the relevant protein sequence[82]. Mass tolerance was set to 10 ppm for MS1's and 20 ppm for MS2's. Met oxidation was set as a variable modification and carbamidomethyl Cys was set as a fixed modification. For samples treated with PNGaseF, Asn deamidation was added as a variable modification. For most samples, we used the default O-glycan database containing 12 common structures. For analysis of GP1bα, the database was based on the previously determined glycomic profile[8]. Files initially underwent a nonspecific search in order to determine the cleavage specificity of SmE. After the cleavage motif was determined, files generated using only an O-glycoprotease digestion were searched with semi-specific cleavage N-terminal to Ser and Thr and six allowed missed cleavages. Samples treated with trypsin were searched with the same parameters, but also allowed cleavage C-terminal to Arg or Lys. Results were filtered to a q value less than 0.01 and manually validated using Xcalibur software (Thermo Fisher Scientific). Relative abundances were obtained by generating extracted ion chromatograms and determining area under the curve. After abundances were obtained, each file was checked for the presence of the identified species. When a peak with matching retention time and mass was present, the peak was validated and the abundance recorded. The mass spectrometry proteomics data have been deposited to the ProteomeXchange Consortium via the PRIDE partner repository with the dataset identifier PXD039583.

## Manual validation of search results

After filtering, each identification was validated by at least one person before being added to the curated result files (SI Tables 1–8). For each putative glycopeptide, the extracted ion chromatograms, full mass spectra (MS1s), and fragmentation spectra (MS2s) were investigated in XCalibur QualBrowser (Thermo). The MS1 was first used to confirm the precursor mass and chosen isotope was correct. It also allowed us to identify any co-isolated species that could interfere with the MS2s and/ or explain unassigned peaks. The HCD and ET(hc)D fragmentation spectra were then investigated to identify sufficient coverage to make a sequence assignment. When possible, multiple MS2 scans were averaged to obtain a stronger spectrum. For HCD, an initial glyco-peptide identification was confirmed if the presence of the precursor mass without a glycan present (i.e., Y0), along with nearly full coverage of b and y ions without glycosylation. For longer peptides, we required the presence of Y0 and fragments that were expected to be abundant (e.g., N-terminally to Pro, C-terminally to Asp). When the peptide contained a Pro at the C-terminus, the $b_{n-1}$ was considered sufficient. Further, when the sequence contained oxidized Met, the Met loss from the bare mass was considered as representative of the naked peptide mass. We then used electron-based fragmentation MS2 spectra for localization. Here, all plausible localizations were considered, regardless of search result output. We confirmed the presence of fragment ions in ETD or EThcD that were between potential glycosylation sites, if sufficient c/z ions were present then a glycan mass was considered localized.

Other important considerations during manual validation of search results:

- After the initial identification of a particular peptide sequence in a strong spectrum, less stringent conditions were needed if the same peptide occurred with a different glycan structure. We used the stronger fragmentation spectrum to determine characteristic fragmentation masses, and then weaker spectra were assigned based on fragment abundance similarity (akin to manual spectral matching).
- In EThcD spectra, ions that had the glycosylation present on the fragment (i.e., c/z ions) were considered more important for localization than the ones that show the fragment without glycosylation (i.e., b/y ions).
- For peptides with a 138/144 ratio under 1.2, we assumed that all glycans in the spectrum were core 1 structures. That is, if two sites did not have coverage in ET(hc)D but the glycan composition was N2, H2N2, or H2N2A4, it was assigned as two N1, H1N1, or H1N1A2 structures to both sites. This was not the case if there was an oxonium ion present at 407 m/z, which would indicate the presence of 2 HexNAcs in a single glycan structure.
- When multiple analyses were being compared (e.g. different O-glycoprotease digestions) that could be expected to have the same unique glycopeptides, any given assignment was checked across all files at the same time. Localization in one file was assumed in the other file(s) if the scan was too weak to localize and (a) retention time was identical, and (b) no contradictory information existed in the sequence and/or localization.
- The list of results generated by trainees was reviewed by senior members of the lab. Additionally, any unlikely identifications (e.g., uncommon glycans, a single identification of a glycopeptide) were manually annotated by the trainee and submitted to a senior member for approval before they were included in the curated results.

To obtain as many glycopeptides as possible for the glycoproteomic maps throughout the manuscript, further identifications were made beyond search result verification. This allowed for the identification of mutations, uncommon glycan structures, and anything else that did not fall within the search parameters. We performed these analyses in a number of different ways:

- We extracted the HexNAc fingerprint ion (204.0867) from MS2 spectra, which gives the overall intensity of the oxonium ion

throughout the run. Alternatively, we extracted any HCD scans that triggered an ET(hc)D scan. If any of these highly abundant glycosylated species were not identified by the search algorithm, they were manually de novo sequenced.

- Further, if we did not obtain 100% sequence coverage for a mucin domain, peptide sequences were predicted based on the cleavage motif(s) of the enzyme(s) used. Candidate spectra were found by extracting all scans with oxonium ions and expected fragments from the peptide (e.g., Y0 and N-terminal Pro cleavages). These scans were further filtered by subtracting the naked peptide mass from the precursor mass and matching the remaining mass to different combinations of glycan masses. For spectra that had mass differences matching a glycan mass, the sequences were then validated and localized as described above.

- Finally, to assure we obtained the highest number of glycoforms (i.e., peptides with different glycan compositions and/or glycosites), we used a similar extraction technique. For any potential glycopeptide that was poorly scored/identified by the search algorithm, we extracted expected fragments and any spectra with a new glycan composition were then validated and localized as described above. For example, TIAVFT (TIM-4) had a lot of identified glycoforms (>50) because it fragmented well by HCD, whereas the search algorithm only identified one TVRT (TIM-1) unique glycopeptide. Using this technique we found a total of 6 unique glycopeptides from TVRT, all of which were more abundant than the identified one (2xN1), but had worse HCD spectra (i.e., fewer b/y fragment ions) because the glycans were more extended.

## Abundance calculations

To determine individual glycopeptide abundances, we first generated extracted ion chromatograms of monoisotopic masses and calculated areas under the curve (AUCs). To avoid bias toward smaller glycan structures while minimizing the inconvenience of typing multiple isotope masses into XCalibur QualBrowser (Thermo) for each peptide, abundances were calculated using a variable number of isotopes. Only the 12C peak was used for anything under 1600 Da, the 13C was also included up to 2400 Da, and three isotopes were collected over 2400 Da. These values were chosen based on the predicted isotopic distributions for given intact peptide values to allow the majority of the signal to be incorporated. All charge states were included when determining the abundance of a given glycoform. Glycopeptide abundances were taken using the file collected with ETD, even if localization took place using the paired EThcD file. If the protein required a secondary file with PNGaseF treatment in order to identify O-glycosites near N-glycosylation sites (TIM-4, TIM-3). In these cases, the PNGaseF treated file was used to record abundances.

To generate Fig. 4A and determine the most abundant glycan at each site of TIM-3 and TIM-4, we used the above method with additional steps. First, the most abundant core type was determined based on the combined area from each category (Tn, core 1, core 2, or "other"), then the most abundant glycan within the category was chosen to investigate. Thereafter, when a residue was observed with a particular glycan structure, but with different glycans throughout the rest of the peptide (i.e., different glycoforms), the abundances of all peptides containing that glycosite with that glycan were summed to get the value for the "most abundant glycan".

## Manual annotation ("Markups")

Please see Appendix 1 for representative spectra that have been annotated. These "markups" have a cover sheet showing the sequence, assigned localization, fragment ions identified in the spectra (here, HCD calculated sans glycosylation, ETD with glycosylation), the difference from calculated mass, and any additional comments/reasoning/observations.

On the first page, a checkmark indicates that the fragment mass associated with a fragmentation type is present in an included spectrum (b/y for HCD, c/z for ExD). Low-abundance ions were denoted as weak (w) or very weak (vw). Assignments that were ambiguous (e.g., assignable to multiple possible fragment species) were marked with a question mark (?). Neutral losses and fragment ions from the non-standard series were denoted here for each fragmentation type: HCD: water loss (o), ammonia loss (N), a-ion (a), fragment observed with glycosylation attached (*); ETD: a●-ion (a), ETD y-ion (y). EThcD may include a combination of these notations. Internal fragments seen in HCD were shown using a line next to the vertical sequence that covers the relevant amino acid stretch.

Following the cover page is one or more annotated HCD spectra followed by annotated ETD and/or EThcD spectra. Checkmarks on these spectra included the fingerprint ions expected from the assigned glycan structure(s). Sequence-informative ions were labeled with a/b/c/y/z, the fragment number, and the charge. Neutral losses were usually depicted with a line from the originating fragment but may be shown as the fragment assignment and the loss (e.g., b4 + -H$_2$O).

The last page contains more general information that can be helpful for making an assignment. On the left, it displays the total ion current, base peak, elution profile of the anticipated charge state(s), and the MS2 scan times. On the right, zoomed MS1 peaks of the precursor showed the isotopic distribution of the peak, whether there were co-isolated species, and the detected exact mass of the precursor. If multiple peptides with the same overall glycan composition were detected, this page will have labels above the peaks connecting them to their respective spectra, labeled with a number over the peak which is also on the cover page for the assignment. If multiple files were used for an identification, they will be marked in different colors to show where the information was collected.

## Cell culture

HeLa cells (ATCC, CCL-2) were grown in T75 flasks (Falcon, 353136) and maintained at 37 °C, 5% CO$_2$. The cells were cultured in DMEM (Gibco, 11965-092) supplemented with 10% fetal bovine serum (FBS, Sigma, F0926), 1% sodium pyruvate (Gibco, 11360-070), and 1% penicillin/streptomycin (Cytiva, SV30010).

## Western blotting for MUC16 on SmE and StcE-treated cells

HeLa cells were seeded in T25 flasks (Falcon, 353109). The following day, the media was removed and dilutions of StcE or SmE (0, 0.05, 0.5, 5, and 50 nM) in Hank's buffered salt solution (HBSS, Gibco, 24020-117) were added for 60 min. Following treatment, media (1 mL) was collected into tubes containing 0.75 µL of 0.5 M EDTA (Invitrogen, 15575-038) to quench the enzymatic reaction. The samples were then concentrated in a 3 kDa spin filter (Millipore, UFC500324). The cells remaining in the flask were washed with an enzyme-free dissociation buffer containing EDTA (Millipore, S-004-C), lifted, and transferred to tubes. The cells were pelleted and washed with PBS (Gibco, 14190-144) twice, then lysed by boiling in 1x NuPAGE LDS Sample Buffer (Invitrogen, NP0008) supplemented with 25 mM DTT at 95 °C for 5 min. Concentrated supernatants were diluted in 4x sample buffer to a final concentration of 1x. Both cell lysates and supernatants were boiled for 5 min at 95 °C and 30 µL of sample was loaded onto a 4–12% Criterion XT BisTris gel (Bio-Rad, 3450124) which was run in MOPS XT buffer (Bio-Rad, 1610788) at 180 V for 90 min. Proteins were transferred to a 0.2 µm nitrocellulose membrane (Bio-Rad, 1620112) using the Trans-Blot Turbo Transfer System (Bio-Rad), at a constant 2.5 A for 15 min. Total protein was quantified using Revert 700 stain (LI-COR Biosciences, 926-11011) before a 1:1000 dilution of primary antibody (clone X75, Novus Biologicals, NB600-1468) was incubated overnight at 4 °C. A 1:15,0000 IR800 dye-labeled secondary antibody (LI-COR, 926-32210) was used according to manufacturer's instructions for visualization on a LI-COR Odyssey instrument.

## Cell viability assay

HeLa cells (CCL-2, ATCC) were seeded in 24-well plates at approximately 20,000 cells per well in 500 μL of media. After 24 h, SmE and StcE were added at 500, 5, 0.05, and 0 nM. At $t = 0$, 24, 48, 72, 96 h post-treatment, PrestoBlue (Invitrogen, A13261) was added according to the manufacturer's instructions. After 2 h, the supernatant was transferred to a black 96 well plate (Thermo Scientific, 237105) for fluorescent readings on a SPECTRAmax GEMINI spectrofluorometer using an excitation wavelength of 544 nm and an emission wavelength of 585 nm. Results were plotted and statistical significance was assessed by two-wayANOVA in GraphPad Prism.

## Molecular dynamics simulations

Note on Model Scale and Parameter accuracy: Given the size of TIM-3 and TIM-4 glycoproteins, and given the nature of the biophysical questions addressed in this work, all-atom models (as described below) were deemed most appropriate to capture large-scale protein motions while maintaining accuracy. Additionally, given the fact that there currently exist no parameters for coarse-grained O-glycan models, the all-atom scale was the most efficient and only scale capable of modeling TIM-3 and TIM-4 at the time of this work. Additionally, given the importance of hydrogen bonding in glycan dynamics and conformations, implicit solvent was deemed inappropriate for modeling and thus explicit solvent (TIP3 waters) was determined to be required. Finally, the CHARMM36 force field is one of the most well-developed force fields for the simulation of glycans[83].

**Protein construction.** TIM-3 protein model was built from the following component models: X-ray crystal structure of the TIM-3 IgV domain (PDB 7M41)[84], mucin-domain backbone constructed from the BuildPeptide tool in ROSETTA[85] from the FASTA sequence, and the transmembrane tail and cytoplasmic domains modeled with Alpha-Fold (AF Q8TDQ0)[51,52]. TIM-4 protein model was built from the following component models: X-ray crystal structure of the TIM-4 IgV domain (PDB 5F7H)[86], mucin-domain backbone constructed from the BuildPeptide tool in ROSETTA from the FASTA sequence, and the transmembrane tail and cytoplasmic domains modeled with Alpha-Fold (AF Q96H15). Each of these domains were then joined together using psfgen in VMDTools[87]. PropKa3[88] was used to identify appropriate protontation states at pH 7.4: All aspartate and glutamate residues were determined to be deprotonated and negatively charged; histidines were determined to be singly deprotonated and neutral (kept in HSD protonation state); lysines and arginines were determined to be protonated and positively charged; tyrosines were determined to be protonated and neutral.

**Glycosylation.** An FA2 glycan was chosen for the identified N-linked glycan positions on TIM-3 and TIM-4 (N99 & N171, N291, respectively) as that was consistent with the known glycoprofile at these positions, and full characterization of N-linked glycan positions is outside the scope of this current work. For all O-linked glycans, the glycan structure at each position with the highest population, as determined by MS, was chosen and modeled and constructed onto each site using CHARMM-GUI[89].

**Lipid bilayer insertion.** Complete TIM-3 and TIM-4 models were then inserted into lipid bilayer patches with compositions similar to that of mammalian cell membranes (56% POPC, 20% CHL, 11% POPI, 9% POPE, and 4% PSM)[90,91].

**Solvation and neutralization.** Finally, the TIM-3 and TIM-4 systems were embedded into orthorhombic boxes, explicitly solvated with TIP3 water molecules, and neutralized to a concentration of 150 mM of NaCl, resulting in systems of 846,793 and 2,122,863 million atoms, respectively. See Table 1 for a complete system breakdown:

**Table 1 | Compositional breakdown of each structure simulated in this work**

| | TIM-3 | TIM-4 |
|---|---|---|
| Total #Atoms | 846,793 | 2,122,863 |
| #Protein atoms (#AAs) | 4316 (280) | 5336 (355) |
| #Glycan atoms (#segments) | 1275 | 3410 |
| #Water atoms (#molecules) | 781,557 (260,519) | 2,022,120 (674,040) |
| #Na/#Cl atoms (150 mM NaCl) | 830/735 | 2055/1902 |
| Lipid Bilayer Size (Å × Å) | 130 × 130 | 140 × 140 |
| #CHL1 atoms (#segments) | 5032 (68) | 8066 (109) |
| #POPC atoms (#segments) | 29,346 (219) | 43,550 (325) |
| #POPE atoms (#segments) | 12,250 (98) | 18,125 (145) |
| #POPI atoms (#segments) | 7261 (53) | 12,330 (90) |
| #POPS atoms (#segments) | 4191 (33) | 5969 (47) |
| Box Size (Å × Å × Å) | 140.7 × 150.6 × 473.2 | 177.9 × 170.4 × 818.8 |

Lipid bilayer patch size and box size reflect dimensions at $t = 0$ before any simulation was conducted.

**Molecular dynamics (MD) simulations.** All MD simulations were performed with NAMD2.14[92] and CHARMM36m all-atom additive force fields[93–99] on a private supercomputer in the Triton Shared Computing Cluster hosted by the San Diego Supercomputer Center[100]. *Lipid tail minimization and melting:* All atoms except lipid tails were held fixed according to a Lagrangian constraint (i.e. the "fix" command in NAMD), while lipid tails were subjected to 10,000 steps of Steepest Descent minimization. Then, a heating step was performed wherein, with constraints on all atoms except for lipid tails, the system temperature was incrementally increased from 10 K to 310 K for 0.5 ns at 1 fs/step. *Total system minimization and equilibration:* Following lipid tail melting, the Lagrangian constraints were removed from all atoms, but an energetic restraint (1 kcal/mol/Å) was applied to all protein and glycan atoms. The complete system was then subjected to 10,000 steps of Steepest Descent minimization, followed by 0.5 ns of equilibration at 310 K (at a 1 fs timestep). After free total minimization and restrained equilibration, the TIM-3 and TIM-4 structures were branched to perform replicas of the following MD simulation protocols. TIM-3 was branched into 4 replicas and TIM-4 was branched into 3 replicas. *Free equilibration:* Finally, all restraints were removed (thus no restraints or constraints on the system at all) and all atoms were allowed to equilibrate for 0.5 ns at 310 K (1 fs/step). *Production:* A total of 700 ns and 830 ns (2 fs/step) were collected for TIM-3 and -4, respectively, see Table 2 for a breakdown of per replica sampling. MDAnalysis was then used to perform all of the resultant analyses from MD simulations[101,102].

**Persistence length calculations.** The polymer module in MDAnalysis[101,102] was used to calculate the persistence length of TIM-3 and TIM-4 mucin domains by defining a mucin polymer as the N, CA, and C, atoms along the backbone for the following residue selections: TIM-3, residues 133 to 198, TIM-4: residues 137 to 310. Per replica and per TIM protein, persistence length was calculated using the last 1000 frames of simulation. We calculated persistence lengths from each replica trajectory, and thus we have reported the average and standard deviation of persistence lengths calculated for each TIM mucin domain over the three replicas.

**End-to-end distance calculations.** The distances module in MDAnalysis[101,102] was used to calculate the distance in Angstroms (Å) between the center of mass of the last residue of the globular IgV domain and the center of mass of the first residue of the transmembrane helical domain for every frame in each simulation trajectory. For TIM-3 these first and last residues were selected as 133 and 198,

**Table 2 | Complete breakdown of total simulation time for TIM-3 and TIM-4**

|  | TIM-3 (ns) | TIM-4 (ns) |
|---|---|---|
| Rep 1 | 238.8 | 260.9 |
| Rep 2 | 152.2 | 309.9 |
| Rep 3 | 131.3 | 261.0 |
| Rep 4 (TIM-3 only) | 177.1 | – |
| Total | 699.7 | 831.8 |

respectively, and for TIM-4 these residues were 137 and 310, respectively. We then normalized the calculated distances by the total number of protein residues within each mucin domain.

**Bending angle calculations.** To calculate the bending angle, we used MDAnalysis[101,102] to select the center of mass of the first, middle, and final protein residue within each mucin domain. We then used these positions to calculate vectors: one from the middle residue to the first residue of each mucin domain, and one from the middle residue and the last residue of each mucin domain. For TIM-3, the first, middle, and last residues were selected as residues 131, 166, and 202, respectively. For TIM-4, the first, middle, and last residues were selected as residues 135, 225, and 314, respectively. We then calculated the angle between these vectors for each frame for all three replica trajectories. We plotted these results as normalized density histograms.

**Shared structures, trajectories, and input files.** A compressed tar.gz file is included with the supporting information of this work. These shared files include: starting coordinates, final coordinates, NAMD input files, and complete stripped trajectories of TIM-3 and TIM-4 glycoproteins.

**Mucinase structure overlays, molecular docking, and TIM-4 grafting**

Structural comparison and docking were performed using Molecular Operating Environment (MOE) 2020.09. The X-ray crystal structures of ImpA[30,46], AM0627[49,50], ZmpB[30], ZmpC[48], and BT4244[30] were superimposed with the AlphaFold-predicted structures[51,52] of SmE, AM0908, and AM1514 using the catalytic histidine and glutamate residues to guide alignment[45].

Following this, a ligand was generated over several steps for use in docking studies. First, the bisglycosylated ligand cocrystallized with AM0627[50] was placed into the analogous location of the superimposed SmE model structure. The amino acid residues of the peptide were then mutated to match a TIM-4 sequence (Ala189-Val190-Phe191-Thr192*-Thr193*-Ala194, where the asterisk indicates glycosylation) that was found to be cleaved by SmE but not ImpA. The peptide's N terminus was acetylated and its C terminus was N-methylated to better approximate the steric/electronic environment of a substrate. While holding the SmE model structure fixed, the glycopeptide was allowed to preliminarily minimize in the Amber10:EHT forcefield[103] using restraints to ensure contacts were formed between (1) the backbones of the glycopeptide and the beta strand of the active site as well as (2) the GalNAc moiety and the side chains of the conserved residues Trp240 and Asn268.

During our glycoproteomic mapping, we observed high occupancy of H1N1A1 (or smaller fragments thereof) modifying Thr192 at P1 and H2N2A1 (or smaller fragments thereof) modifying Thr193 at P1′ (Supplementary Fig. 18). Similar to previous work[17,104], the P1 glycan was generated by grafting a sialic acid residue onto the 3-OH of the Gal moiety, using the glycan bound to the GspB Siglec domain (PDB 5IUC) as a template[105]. The P1′ glycan was similarly generated by grafting the remaining GalNAc, Gal, and Sia residues onto the 6-OH of the GalNAc moiety branching from P1′ using the glycan present on PSGL-1 in its cocrystal structure with P-selectin (PDB 1G1S)[106].

Following this, the resulting glycopeptide underwent conformational search, holding all atoms except for the newly-grafted glycans fixed, using LowModeMD to generate a library of over 1300 different conformations of these sugar residues[107]. Each conformer underwent virtual screen, holding the mutant SmED245A enzyme (replacing the catalytic residue to facilitate docking) rigid while allowing the glycopeptide ligand to move freely. The ligand-enzyme complexes with the top 100 docking scores were then used in induced fit docking, keeping both the ligand and the enzyme free. This stepwise search, screen, and docking process allowed us to thoroughly explore conformational space of the ligand while minimizing computational resources.

Following this, the docked ligand was then grafted together with TIM-4 fragments from dynamics simulations to generate larger glycopeptide ligands. First, data from the TIM-4 simulation were analyzed to identify frames containing structures that have Ramachandran angles for Val190 and Phe191 that were within ±15° of those in the docked structure (Val190: $\varphi = -140°$, $\psi = 152°$; Phe191: $\varphi = -136°$, $\psi = 102°$). Corresponding fragments (Pro175-Phe191) of ten different structures were manually superimposed on the docked structure, using Val190 and Phe191 to guide placement. Finally, the docked ligand (Val190-Ala194) was grafted onto each fragment (Pro175-Ala189) to generate ten larger glycopeptides used to identify the potential for interaction with the mucin-binding module of SmE.

**Cloning and constructs**

As previously described[74], the plasmid used for expressing the extracellular region of TIM-3 in mammalian cells included human TIM-3 (natural signal sequence with amino acids S1-R179), which was cloned into pcDNA3.1(+) with the addition of a hexa-histidine tag at the C-terminus of the coding sequence by Gibson assembly. pCDEF3-hTIM3, a generous gift from the laboratory of Dr. Lawrence Kane (Addgene plasmid #49212), was used as the insert following correction of the natural variant L119 to Arg by site-directed mutagenesis. The vector used for expressing Gal-9, pET-ssG9 (Riken BioResource Center, cat# RDB15282), contained a modified version of human galectin-9 with a shortened linker and a single amino acid substitution (A > P) for enhanced protein stability and solubility in the pET11a expression vector[73]. Gal-9[R65D] and Gal-9[R200D] were generated with the QuikChange II XL Site-Directed Mutagenesis Kit according to the manufacturer's (Agilent Technologies) instructions. The following forward (F) and reverse (R) primers were designed following the primer design guidelines specified by the kit instructions and were used to generate cDNAs for mutated variants of Gal-9:

Gal-9[R65D]-F (CCTTCCACTTCAACCCTGACTTTGAAGATGGAGGGT ACG)

Gal-9[R65D]-R (CGTACCCTCCATCTTCAAAGTCAGGGTTGAAGTGGA AGG)

Gal-9[R200D]-F (GCCTTCCACCTGAACCCCGATTTTGATGAGAATGCT GTG)

Gal-9[R200D]-R (CACAGCATTCTCATCAAAATCGGGGTTCAGGTGGA AGGC)

All plasmids, including the mutated variants generated by site-directed mutagenesis, were confirmed by DNA sequencing before use.

**Expression and purification of recombinant Gal-9**

*E. coli* BL21(DE3) Codon Plus RIPL cells were used to express recombinant Gal-9 (wildtype and mutant) was carried out as described[108]. *E. coli* (220 mL) transformed with and selected for expression of the pET11a vector containing Gal-9 (WT, R65D, or R200D) were cultured to OD600 = 0.6. Protein expression was induced with 0.1 mM isopropyl-β-D-thiogalactopyranoside (IPTG) at 20 °C overnight (16 h). After overnight induction, cells were pelleted and resuspended in 36 ml of 10 mM Tris-HCl (pH 7.5), 0.5 M NaCl, 1 mM PMSF (phenylmethylsulfonyl fluoride). The cell suspension was sonicated on ice with an ultrasonic processor (Cole-Parmer) for 6 cycles of 2 min sonication in 1 s on/1 s off

bursts at 20% amplitude, with 1 min rest intervals between cycles. Following sonication, lysate was mixed with 10% (w/v) Triton X-100 (4 mL) by gentle stirring for 30 min at 4 °C. Lysate was clarified by centrifugation at 18,000 xg, 4 °C for 30 min. Recombinant proteins were recovered by batch-wise absorption on 2 ml of α-lactose-agarose gel (Sigma-Aldrich). The gel was packed into a Kontes column by gravity flow, washed with 20 gel-bed volumes of 20 mM Tris-HCl (pH 7.5), 0.15 M NaCl, eluted with 1 gel-bed volume of 20 mM Tris-HCl (pH 7.5), 0.15 M NaCl, 0.2 M lactose. Peak fractions were then dialyzed against PBS, pH 7.5 (phosphate buffered saline). The dialysate was centrifuged at 25,000$g$ for 20 min to remove minor amounts of precipitants. The resulting supernatant was then desalted into 10 mM HEPES pH 7.6, 0.15 M NaCl using a Zeba™ Spin Desalting Column, 7 K molecular weight cut-off and further sterilized by filtration (0.22 μm). The protein concentration was determined by measuring the UV absorbance at 280 nm and converting the absorbance into protein concentration using the Beer-Lambert law, using an estimated extinction coefficient of 28420 M⁻¹cm⁻¹. The purified proteins were stored at 4˚C until use.

### Expression and purification of recombinant TIM-3

As previously described[74], TIM-3 was expressed and purified in Expi293 cells, according to the Expi293 Expression System manual. Cells were seeded to $3.0 \times 10^6$ cells/mL and then were transfected with a mixture of DNA vector (pcDNA-TIM-3), Expifectamine reagent (ThermoFisher Scientific), and Opti-MEM. After 4–6 days, culture was harvested by centrifugation, and culture supernatant was dialyzed against 10 mM HEPES pH 7.6, 0.15 M NaCl (4 times the culture volume). Nickel affinity chromatography, ion exchange chromatography, and size exclusion chromatography were then used to purify the diafiltered supernatant. The protein sample was sterilized by filtration (0.22 μm). A NanoDrop spectrophotometer was used to detect absorbance at 280 nm, and protein concentration was determined using an estimated extinction coefficient (25815 M⁻¹cm⁻¹).

### Chemical crosslinking

Purified TIM-3 extracellular region (4 μM) was incubated alone or with Gal-9^WT (0.4, 2, or 4 μM) with or without disuccinimidyl suberate (DSS; 37.5-fold molar excess, 300 μM) for 30 min at 25 °C. Samples were then mixed with NuPAGE™ LDS Sample Buffer (4X) (Thermo-Fisher Scientific) to a final concentration of 1X and with dithiothreitol (DTT) to a final concentration of 50 mM and were boiled at 95 °C for 2 min to quench the reaction. Samples were then run in 4–12% NuPAGE gels and were analyzed by Coomassie staining or were transferred to nitrocellulose membranes for analysis by Western blotting. Coomassie stained gels were imaged with a Gel Doc™ EZ imager (Bio-Rad). Nitrocellulose membranes were blocked with 4% bovine serum albumin in Tris-buffered saline plus Tween20 (TBS-T), probed with a 1:2000 dilution of polyclonal primary α-TIM-3 (R&D Systems, catalog #AF2365) for 1 h or overnight, and then probed with 1:1000 dilution of HRP-tagged goat secondary antibody (R&D Systems, HAF109) for 1 h. SuperSignal Western Pico PLUS Chemiluminescent Substrate was used for detection (ThermoFisher Scientific), and chemiluminescent signal visualized using a Kodak Image Station (Kodak Scientific).

### Vesicle preparation

Dioleoylphosphatidylcholine (DOPC) and dioleoylphosphatidylserine (DOPS) were purchased from Avanti Polar Lipids in chloroform solution. Lipid solutions were combined at the appropriate molar ratios of DOPS/DOPC in a glass vial. Chloroform was dried first by nitrogen gas, then by exposure to vacuum. Following rehydration with 0.5 mL 10 mM HEPES pH 7.6, 150 mM NaCl, lipid solutions were mixed by vortexing and were subjected to at least 10 cycles of liquid nitrogen freezing and thawing in a sonicating water bath to generate unilamellar vesicles. After freeze/thaw cycles, unilamellar vesicles were stored at −20 °C. Sonicated vesicles were thawed and extruded before use using an Avanti Mini Extruder (100 nm filter membrane) according to the manufacturer's instructions.

### Surface plasmon resonance studies

Surface plasmon resonance (SPR) analysis of TIM-3–PS interactions in the presence of Gal-9 was performed with a Biacore 3000 instrument, as described[74]. Lipid vesicles containing 100% DOPC or 20% DOPS were immobilized by flowing extruded lipid vesicles across an L1 sensorchip (Cytiva). Protein solutions containing a fixed concentration of TIM-3 (6 μM) with 1 mM CaCl2 and two-fold serially diluted wildtype or mutant ssGal9 flowed across lipids immobilized on the sensorchip. Resonance units detected by the Biacore 3000 were corrected for background (100% DOPC) binding and were plotted as a function of Gal-9 concentration. The dissociation kinetics were determined with GraphPad Prism using a nonlinear regression model with the dissociation one phase exponential decay equation $Y = (Y0-NS)*exp(-K*X) + NS$ where Y0 is the binding at time zero, NS is nonspecific binding at infinite times, and K is equivalent to the rate constant for the dissociation of the protein–ligand complex, $k_{off}$.

### Reporting summary

Further information on research design is available in the Nature Portfolio Reporting Summary linked to this article.

## Data availability

The mass spectrometry proteomics data and search algorithm outputs have been deposited to the ProteomeXchange Consortium via the PRIDE partner repository with the dataset identifier PXD039583. All other data is provided in the Supplementary Data files as indicated in the text. Source data are provided with this paper.

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

## Acknowledgements

The authors thank Jeffrey Shabanowitz for his technical expertise and thoughtful conversations during the preparation of this manuscript. We also would like to acknowledge the Keck Biophysical Resource for their assistance in cell viability assays. J.C. is supported by an NSF GRFP (DGE-2139841); A.D.S. is supported by the National Institutes of Health Chemical Biology Training Grant (T32 GM067543); K.E.M. and T.M.L. are supported by Yale Endowed Postdoctoral Fellowships in the Biological Sciences. M.A.R. and F.L.K. acknowledge the San Diego Supercomputer at UCSD for providing HPC resources that have contributed to the research results reported within this paper. R.E.A. acknowledges support from NIH GM132826, NSF RAPID MCB-2032054, a UC San Diego Moores Cancer Center 2020 SARS-COV-2 seed grant, and U19-AI171954 from NIAID. M.A.R. is supported by NIH T32 EB009380. This work was also supported by a Sarafan ChEM-H Physician-Scientist fellowship (to M.A.H.), the Stanford Maternal & Child Health Research Institute Instructor K Award Support Program (to M.A.H.), a National Blood Foundation Early Career Scientific Research Grant (to M.A.H.), a NIH NHBLI Pathway to Independence award (1K99HL156029-01 to M.A.H.), and NIH grant R01CA200423 (to C.R.B.). D.J.S. was supported by a US NSF Graduate Research Fellowship and Stanford Graduate Fellowship M.J.F. was supported by a University of Redlands Faculty Research Grant. C.J.M., A.L., and M.A.L. were supported by NIGMS grant R35-GM122485 (to M.A.L.). S.A.M. is supported by the Yale Science Development Fund and a NIGMS R35-GM147039.

## Author contributions

S.A.M., R.E.A., M.A.L., and C.R.B. advised the project and oversaw experiments. S.A.M and K.E.M. designed mass spectrometry experiments. J.C., A.D.S., K.E.M., T.M.L., and D.I. performed mass spectrometry and cell biology experiments. J.C., A.D.S., K.E.M., T.M.L, D.I., A.S.B., V.A., and C.K. analyzed mass spectrometry data. M.J.F. performed molecular docking experiments and analyzed associated data. M.A.H, D.J.S., and K.H.T. provided materials and plasmids. M.A.R., F.L.K., and R.E.A. designed all MD simulations. M.A.R. constructed TIM-3, TIM-4 models and conducted MD simulations. M.A.R. and F.L.K. conducted analysis from MD simulations. C.M.S. and A.L. performed functional assays, advised by M.A.L. S.A.M., J.C., A.D.S., T.M.L., C.M.S., A.L., K.E.M., M.J.F., M.A.R., and F.L.K. wrote the manuscript with additional input from all authors.

## Competing interests

F.L.K. is a consultant for Protein Evolution, Inc. M.A.H. received consulting fees from Dova Pharmaceuticals, Janssen Pharmaceuticals, and Sonder Capital. C.R.B. is a co-founder and scientific advisory board member of Lycia Therapeutics, Palleon Pharmaceuticals, Enable Bioscience, Redwood Biosciences (a subsidiary of Catalent) OliLux Bio, Grace Science LLC, and InterVenn Biosciences. S.A.M. is a consultant for InterVenn Biosciences and Arkuda Therapeutics. S.A.M., D.J.S., and C.R.B. are coinventors on a Stanford nonprovisional utility patent application that has been filed and is pending in the US (number US20220003777) related to the use of mucinases for mass spectrometry analysis of mucin-domain glycoproteins. The remaining co-authors have no conflicts of interest to disclose.
