## [Peer Review File · Nature Communications]

REVIEWER COMMENTS

Reviewer #1 (Remarks to the Author):

Mucins represent a very important class of biological molecules that are incredibly challenging to study given the presence of extensive glycosylation, the inherent extended nature of their 3D structures and their location on cell surfaces. The present work makes significant advances in the analytical tools and techniques to investigate this class of molecules. As part of the investigation, motivated on the inability to obtain experimental structures for the studied molecules, molecular dynamics simulations are employed leveraging the information on the determined glycosylation pattern from the experimental portion of the study from which preliminary structural and dynamical information on the mucins is reported. Overall this is an important study that clearly advances the field.

Given my expertise in computational methods my detailed review focuses on that portion of the manuscript.

The MD simulation methods are well documented and generally appropriate given the size of the mucin molecules being studied. However, the presented results are clearly not converged. That said, to obtain converged results would require a truly heroic effort including the application of enhanced sampling methods, potential of mean force calculations, and microsecond simulations of these very large systems. Thus, while the results extracted from the simulations are of utility to understand differences between two TIM mucins the authors need to clearly state that the observations concerning the persistence length and bending angles are not fully converged and should be considered preliminary in nature.

In SI computational methods correct "(list out the N-linked glycan positions)" text with the requested data.... and "CHARMM64m"

Reviewer #3 (Remarks to the Author):

Chongsaritsinsuk et al presents a new and powerful technique (a workflow) based on the potential of mucinase SmE to enable a deep characterization of the mucinome of a key family of immune checkpoint inhibitors, the TIM family. The toolkit that was developed is definitely a step forward towards a comprehensive characterization of mucin-domain glycoproteins.

The group has been carrying out an interesting methodological development. in previous studies. focused on the mapping of the mucinome of different mucin-domain glycoproteins, which is a challenging approach given the dense glycosylation characteristic of mucin domains. In this study, they demonstrated that TIM-3 has fewer O-glycosites when compared to TIM-1 and -4, associated with shorter persistence length and higher flexibility in comparison with TIM-4.

The manuscript is pretty much based in the molecular characterization of mucin domain of TIM family highlighting that predicted glycosylation sites in the TIM family (currently described in literature) could have significant limitations as it does not provide a clear insight on the true glycoproteomic landscape of the mucin domain glycoproteins.

Despite this major methodological advance with a broad range of potential applications in other glycoproteins harboring mucin domains, the major concern of this ms is the fact that it is not clear how this differential glycoproteome (mucinome) directly affects the activity and function of TIM molecules:

- What is the impact of this differential mucinome in TIM protein ligand binding and intracellular signaling? What is the impact of SmE clearance of mucins from TIM3 in its binding affinity to its ligand (e.g. Galectin 9; CEACAM1; HMGB1). What is the functional implications of having fewer O-glycosites found to be present in TIM-3 in comparison with TIM-1 or -4 in terms of interaction with its ligands?

The inclusion of functional assays will definitely improve the potential application of having a clear mucinome characterization in terms of structure-function relationship, opening new avenues for leveraging the glycoproteomic landscape of TIM family to improve cancer immunotherapy.

Reviewer #2 (Remarks to the Author):

Please see the attached file

In the current work Chongsaritsinsuk and coworkers characterized a new bacterial enzyme, a mucinase (SmE from *Serratia marcescens*) and after assessing its substrate specificity they used this enzyme to analyze the glycosylation landscape of the TIM family (TIM1, 3 and 4) followed by molecular dynamics simulations of the densely glycosylated TIM4 and more sparsely glycosylated TIM3 in order to better understand how glycosylation effects structural features of these proteins.

As molecular dynamics simulation is outside my area of expertise, I only cover the first part of the manuscript, e.g. mass spectrometry-based O-glycosylation analysis. The authors used model O-glycoproteins to demonstrate that SmE is indeed a mucinase, e.g. it effectively cleaves densely O-glycosylated proteins but is less effective towards non-mucin O-glycoproteins. Following this the authors determined the substrate specificity of SmE using five mucin-domain containing glycoproteins using LC-MS/MS applying HCD and ETxD activation and showed that SmE is more permissive towards different glycan structures and also effectively cleaves between two glycosylated residues as opposed to other O-glycoproteinases including OgpA and ImpA, concluding that currently it is the best enzyme to characterize densely O-glycosylated proteins. With this knowledge in hand the authors explored the glycosylation of TIM1, TIM3 and TIM4, and using the most abundant glycoform at each glycosylation site they assembled the most abundant protein glycoform of TIM3 and TIM4 to investigate the structural and dynamical impact of glycosylation using MD simulations. The authors indeed developed a powerful workflow to study the structural dynamics of mucins.

My major concern regarding the results is the validity of the data interpretation. Unfortunately, the authors applied too permissive acceptance parameters for O-glycopeptide PSMs. Allowing nonspecific cleavages in combination with multiple modifications/peptide and many missed cleavage sites results in many possible (glyco)peptide candidates for the same precursor (even if a single protein is analyzed) and these candidates might differ not only in the modification sites but also in the glycan structures or the peptide sequence. I randomly picked two level1 (indicating unambiguous modification site assignments on all reported sites within the peptide, including fidelity of the assigned glycan structure(s)) PSMs assigned to O-glycopeptides carrying glycans with blood group antigens and found that both assignments are incorrect on all possible levels: the peptide sequence, the glycan compositions and the modifications sites were all incorrect as illustrated with the annotated spectra below.

Therefore, the authors need to adjust the acceptance parameters of the glycopeptide identifications and reanalyze the data to assess the specificity of the O-glycoproteinases covered. I am well aware that currently there is no reliable method to estimate the FDR for glycopeptide identifications. This issue was discussed in detail in a recent paper published in MCP (doi: 10.1016/j.mcpro.2022.100439). Probably the strict requirement of the presence of the Y_0 ion –confirming the molecular weight of the peptide- in the HCD data would be a fairly good starting point. Unfortunately, currently there is no software that applies this requirement including O-Pair used in the current study. Therefore, the next best and the simplest way to filter out the majority of the misidentifications would probably be an additional, strict score-based thresholding (in my experience PSMs with scores $< \sim 20$ are typically unreliable) in combination with removing all non-level1 PSMs.

Regarding that results illustrated in Figures 1-4 (and also SFigures 6-9) are based on O-glycopeptide identifications, reanalyzing the data is a must to reliably justify the findings of the authors. In the revised version of the manuscript the authors should clearly describe and justify their choice of acceptance parameters for glycopeptide identifications.

Original assignment: S(N2H2)LQDRGEGKVAT(N1H2)T(N1)VIS(N1H1F1)K, level1, score: 9.24 (m/z: 940.9305)
Correct assignment: NPN*ATSSSSQDPESLQDRGEGKVAT(N1H1A1)TVISK, or NPN*ATSSSSQDPESLQDRGEGKVATT(N1H1A1)VISK
 (m/z: 940.9368, deamidated N-terminal peptide of the processed protein with 3 missed tryptic sites)

220820_KEM_C1Inh_Trypsin_EThcD #13906 RT: 36.76 AV: 1 NL: 2.10E6
 T: FTMS + c NSI Full ms [300.0000-1500.0000]

220820_KEM_C1Inh_Trypsin_EThcD #13923 RT: 36.79 AV: 1 NL: 8.89E4
 T: FTMS + c NSI d Full ms2 941.1881@hcd29.00 [120.0000-2000.0000]

220820_KEM_C1Inh_Trypsin_EThcD #13925 RT: 36.80 AV: 1 NL: 7.09E4
 T: FTMS + c NSI d sa Full ms2 941.1881@etd24.32 941.1881@hcd15.00 [156.0000-2000.0000]

Original assignment: S(N2H2A1F1)T(N2H2)KKT(N1H1F1)IPELDQPPK; level1, score: 9.1 (m/z: 957.9315)
More plausible: T(N2H2A2)KKT(N2H2A1)IPELDQPPK or T(N2H2A)KKT(N2H2A2)IPELDQPPK (m/z: 957.9224)

220828_KEM_GP1b_SmE_ETD #12953 RT: 35.08 AV: 1 NL: 6.18E6
 T: FTMS + c NSI Full ms [300.0000-1500.0000]

220828_KEM_GP1b_SmE_ETD #12980 RT: 35.14 AV: 1 NL: 1.88E6
 T: FTMS + c NSI d Full ms2 958.1743@hcd29.00 [120.0000-2000.0000]

220828_KEM_GP1b_SmE_ETD #12982 RT: 35.14 AV: 1 NL: 2.37E5
 T: FTMS + c NSI d Full ms2 958.1743@etd24.32 [156.0000-2000.0000]

In addition, the authors claimed that the O-glycopeptide assignments were validated manually (supplementary data, p24: „Results were filtered to a q value less than 0.01 and manually validated using Xcalibur software (Thermo Fisher Scientific)”). What did they mean exactly? Tens of thousands of PSMs clearly cannot be inspected manually. However, I think that the glycosylation profile of each inspected protein should be proven by annotated, true level1 identifications for each and every glycoform – e.g. the authors should demonstrate that the glycosylation sites, and the glycans reported at these sites are indeed verified (see. „Unique glycan site combo” sheet of the supplementary tables).

Minor remarks:

1. On page 3, the authors say „To determine SmE’s mucin selectivity, we digested glycoproteins with and without mucin domains. SmE preferentially cleaved the mucin proteins C1 esterase inhibitor (C1-Inh), CD43, and TIM-1, whereas it did not significantly cleave the non-mucin glycoproteins fibronectin and fetuin (**Figure S2**).” – From Figure S2 it is not evident that CD43 is indeed digested. Please clarify.
2. Figure 1B, P1 and P1’ pie chart – what percentage of the peptides were actually glycosylated at their respective termini? In the samples that were not treated with trypsin, did the authors identify any non-glycosylated peptides? If yes, please include the respective data. (Please reflect after reassessing the data.) Which PSMs were included in the pie charts? – the number of glycopeptides listed in the O-Pair outputs (repository) is much higher than 587 (for P1’).
3. It is not clear how exactly the authors assembled the major glycoforms for MD simulation. How were different glycoforms of the same molecular weight treated? These might or might not separate during reversed phase separation. For example, level1 PSMs were assigned to three SmE-generated glycoforms of the SADTVLL peptide of TIM4 with the total H2N2A1 glycan composition: SADT[H2N2A1]VLL, S[H2N2A1]ADTVLL, and S[H1N1]ADT[H1N1A1]VLL. These isoforms (if these are present at all, as all higher-scoring PSMs belong to the S[H1N1]ADT[H1N1A1]VLL isoform) coelute:

Moreover, many glycosylation sites might be covered by multiple peptides. How was this issue treated?

4. Please include source information for Figure S16 (file, scan#). Please also include the respective annotated HCD spectrum.
5. Appending an explanatory TOC to the supplementary data, and also to the data in the repository would be extremely useful. For example, what do modification sites highlighted in purple („implied”) in the „unique mod sites” sheet stand for? What is the difference between different comparison psmtsv files like oglyco_C1-INH_Comparison_1, oglyco_C1-INH_Comparison_2 and oglyco_C1-INH_Comparison_3? Why is the „Most common per site” tab empty in Supplementary data 412631_0_data_set_7320029_rpmqy7.xlsx? Please rename the supplementary files to reflect their contents/include a TOC file for all supplementary files.

Response to Reviewers

Reviewer #1:

Mucins represent a very important class of biological molecules that are incredibly challenging to study given the presence of extensive glycosylation, the inherent extended nature of their 3D structures and their location on cell surfaces. The present work makes significant advances in the analytical tools and techniques to investigate this class of molecules. As part of the investigation, motivated on the inability to obtain experimental structures for the studied molecules, molecular dynamics simulations are employed leveraging the information on the determined glycosylation pattern from the experimental portion of the study from which preliminary structural and dynamical information on the mucins is reported. Overall, this is an important study that clearly advances the field.

We thank the reviewer for their kind words and for the time spent reviewing our manuscript.

Given my expertise in computational methods my detailed review focuses on that portion of the manuscript.

The MD simulation methods are well documented and generally appropriate given the size of the mucin molecules being studied. However, the presented results are clearly not converged. That said, to obtain converged results would require a truly heroic effort including the application of enhanced sampling methods, potential of mean force calculations, and microsecond simulations of these very large systems. Thus, while the results extracted from the simulations are of utility to understand differences between two TIM mucins the authors need to clearly state that the observations concerning the persistence length and bending angles are not fully converged and should be considered preliminary in nature.

We appreciate the reviewer's comment and have thus added the following statement:

"It is important to note that, although the results extracted from the simulations suggest differences between the two proteins, our observations concerning the persistence length and bending angles are not fully converged due to limitations on time and computing power, so should be considered preliminary in nature."

In SI computational methods correct "(list out the N-linked glycan positions)" text with the requested data.... and "CHARMM64m"

We apologize for this oversight and have corrected the text with the following:

Glycosylation: An FA2 glycan was chosen for the identified N-linked glycan positions on TIM-3 and TIM-4 (N99 & N171, N291, respectively) as that was consistent with the known glycoprofile at these positions, and full characterization of N-linked glycan positions is outside the scope of this current work.

Molecular Dynamics (MD) Simulations: All MD simulations were performed with NAMD2.14²⁰ and CHARMM36m all-atom additive force fields²¹⁻²⁷ on a private supercomputer in the Triton Shared Computing Cluster hosted by the San Diego Supercomputer Center.²⁸

Reviewer 2:

In the current work Chongsaritsinsuk and coworkers characterized a new bacterial enzyme, a mucinase (SmE from *Serratia marcescens*) and after assessing its substrate specificity they used this enzyme to analyze the glycosylation landscape of the TIM family (TIM1, 3 and 4) followed by molecular dynamics simulations of the densely glycosylated TIM4 and more sparsely glycosylated TIM3 in order to better understand how glycosylation effects structural features of these proteins. As molecular dynamics simulation is outside my area of expertise, I only cover the first part of the manuscript, e.g. mass spectrometry-based O-glycosylation analysis. The authors used model O-glycoproteins to demonstrate that SmE is indeed a mucinase, e.g. it effectively cleaves densely O-glycosylated proteins but is less effective towards non-mucin O-glycoproteins. Following this the authors determined the substrate specificity of SmE using five mucin-domain containing glycoproteins using LC-MS/MS applying HCD and ETxD activation and showed that SmE is more permissive towards different glycan structures and also effectively cleaves between two glycosylated residues as opposed to other O-glycoproteinases

including OgpA and ImpA, concluding that currently it is the best enzyme to characterize densely O-glycosylated proteins. With this knowledge in hand the authors explored the glycosylation of TIM1, TIM3 and TIM4, and using the most abundant glycoform at each glycosylation site they assembled the most abundant protein glycoform of TIM3 and TIM4 to investigate the structural and dynamical impact of glycosylation using MD simulations. The authors indeed developed a powerful workflow to study the structural dynamics of mucins.

We thank the reviewer for their time spent reviewing the data and for their commitment to correct glycopeptide spectral analysis.

My major concern regarding the results is the validity of the data interpretation. Unfortunately, the authors applied too permissive acceptance parameters for O-glycopeptide PSMs. Allowing nonspecific cleavages in combination with multiple modifications/peptide and many missed cleavage sites results in many possible (glyco)peptide candidates for the same precursor (even if a single protein is analyzed) and these candidates might differ not only in the modification sites but also in the glycan structures or the peptide sequence.

We purposefully allowed permissive acceptance parameters in order to identify as many possible O-GSMs as we can. However, we manually validate everything, so only the glycopeptide sequences/sites that we considered correct are actually included in our Supplemental Datasets 1-8.

I randomly picked two level1 (indicating unambiguous modification site assignments on all reported sites within the peptide, including fidelity of the assigned glycan structure(s)) PSMs assigned to O-glycopeptides carrying glycans with blood group antigens and found that both assignments are incorrect on all possible levels: the peptide sequence, the glycan compositions and the modifications sites were all uncorrect as illustrated with the annotated spectra below.

We also agree that these two glycopeptides were incorrect as annotated by the search algorithms. We would kindly ask the reviewer to observe that in our curated Datasets (SI Tables 1-8), these peptides were not listed as validated.

Therefore, the authors need to adjust the acceptance parameters of the glycopeptide identifications and reanalyze the data to assess the specificity of the O-glycoproteases covered. I am well aware that currently there is no reliable method to estimate the FDR for glycopeptide identifications. This issue was discussed in detail in a recent paper published in MCP (doi: 10.1016/j.mcpro.2022.100439).

The acceptance parameters are detailed below and in the SI Methods, including a full description of the validation methods and hand-annotated spectra. We did not rely on scoring, software localization, and/or FDR assignments.

Probably the strict requirement of the presence of the Y0 ion –confirming the molecular weight of the peptide- in the HCD data would be a fairly good starting point. Unfortunately, currently there is no software that applies this requirement including O-Pair used in the current study. Therefore, the next best and the simplest way to filter out the majority of the misidentifications would probably be an additional, strict score-based thresholding (in my experience PSMs with scores $< \sim 20$ are typically unreliable) in combination with removing all non-level1 PSMs.

As described above, we did not rely on scoring or filtering in hand validation of our glycopeptides. However, as detailed below and in the SI Methods, we did consider the presence of the Y0 ion as an indication that an O-glycopeptide was present, along with several other parameters.

Regarding that results illustrated in Figures 1-4 (and also SFigures 6-9) are based on O-glycopeptide identifications, reanalyzing the data is a must to reliably justify the findings of the authors. In the revised version of the manuscript the authors should clearly describe and justify their choice of acceptance parameters for glycopeptide identifications.

We respectfully disagree with the reviewer and hope that our detailed description of our manual analysis, along with the hundreds of hand annotated glycopeptide spectra, will convince him/her that our data analysis is extremely thorough.

Original assignment: S(N2H2)LQDRGEGKVAT(N1H2)T(N1)VIS(N1H1F1)K, level1, score: 9.24 (m/z: 940.9305)
Correct assignment: NPN*ATSSSSQDPESLQDRGEGKVAT(N1H1A1)TVISK, or NPN*ATSSSSQDPESLQDRGEGKVATT(N1H1A1)VISK
(m/z: 940.9368, deamidated N-terminal peptide of the processed protein with 3 missed tryptic sites)

220820_KEM_C1Inh_Trypsin_EThcD #13906 RT: 36.76 AV: 1 NL: 2.10E6
T: FTMS + c NSI Full ms [300.0000-1500.0000]

220820_KEM_C1Inh_Trypsin_EThcD #13923 RT: 36.79 AV: 1 NL: 8.89E4
T: FTMS + c NSI d Full ms2 941.1881@hcd29.00 [120.0000-2000.0000]

220820_KEM_C1Inh_Trypsin_EThcD #13925 RT: 36.80 AV: 1 NL: 7.09E4
T: FTMS + c NSI d sa Full ms2 941.1881@etd24.32 941.1881@hcd15.00 [156.0000-2000.0000]

Original assignment: S(N2H2A1F1)T(N2H2)KKT(N1H1F1)IPELDQPPK; level1, score: 9.1 (m/z: 957.9315)
More plausible: T(N2H2A2)KKT(N2H2A1)IPELDQPPK or T(N2H2A)KKT(N2H2A2)IPELDQPPK (m/z: 957.9224)

220828_KEM_GP1b_SmE_ETD #12953 RT: 35.08 AV: 1 NL: 6.18E6
T: FTMS + c NSI Full ms [300.0000-1500.0000]

220828_KEM_GP1b_SmE_ETD #12980 RT: 35.14 AV: 1 NL: 1.88E6
T: FTMS + c NSI d Full ms2 958.1743@hcd29.00 [120.0000-2000.0000]

220828_KEM_GP1b_SmE_ETD #12982 RT: 35.14 AV: 1 NL: 2.37E5
T: FTMS + c NSI d Full ms2 958.1743@etd24.32 [156.0000-2000.0000]

As mentioned above, the two glycopeptides that the reviewer has described as “Original Assignments” are **not** included in the Supplemental Tables 1-8, as we also did not agree with the search output. Further, for the second peptide shown above, we agreed with the reviewer’s assessment and the glycopeptides they assigned are found in the Supplemental Tables (as there were two chromatographic peaks/glycoforms for this peptide). The raw output from all search results were provided in PRIDE for the reviewers to observe the GSMs that were identified; however, all of these were manually validated as evidenced by the SI Tables and the attached appendix with hundreds of annotated glycopeptides.

In addition, the authors claimed that the O-glycopeptide assignments were validated manually (supplementary data, p24: „Results were filtered to a q value less than 0.01 and manually validated using Xcalibur software (Thermo Fisher Scientific)”). What did they mean exactly?

A detailed description of our manual analysis is described below and has now been included in the supplemental methods of the manuscript. The q value is regarding O-Pair outputs; the developers recommend initially filtering the data to less than 0.01.

Tens of thousands of PSMs clearly cannot be inspected manually. However, I think that the glycosylation profile of each inspected protein should be proven by annotated, true level1 identifications for each and every glycoform – e.g. the authors should demonstrate that the glycosylation sites, and the glycans reported at these sites are indeed verified (see. „Unique glycan site combo” sheet of the supplementary tables).

On the contrary, we **did** manually validate tens of thousands of GSMs; this took us hundreds of hours of data analysis and nearly 8 people in the laboratory to complete. In order to demonstrate that this was, in fact, the case, we have provided the reviewer with hundreds of hand-annotated glycopeptides from TIM-4 and TIM-3 (Appendix 1). We hope that this, along with our detailed description of our analysis process, serves as sufficient evidence that we are incredibly thorough and careful when reporting glycoproteomic data.

To help describe exactly how the data analysis was performed, we have included the following information in the SI Methods:

Manual validation of search results

After filtering, each identification was validated by at least one person before being added to the curated result files (SI Tables 1-8). For each putative glycopeptide, the extracted ion chromatograms, full mass spectra (MS1s), and fragmentation spectra (MS2s) were investigated in XCalibur QualBrowser (Thermo). The MS1 was first used to confirm the precursor mass and chosen isotope was correct. It also allowed us to identify any co-isolated species that could interfere with the MS2s and/or explain unassigned peaks. The HCD and ET(hc)D fragmentation spectra were then investigated to identify sufficient coverage to make a sequence assignment. When possible, multiple MS2 scans were averaged to obtain a stronger spectrum. For HCD, an initial glycopeptide identification was confirmed if the presence of the precursor mass without a glycan present (i.e., Y0), along with nearly full coverage of b and y ions without glycosylation. For longer peptides, we required the presence of Y0 and fragments that were expected to be abundant (e.g., N-terminally to Pro, C-terminally to Asp). When the peptide contained a Pro at the C-terminus, the bn-1 was considered sufficient. Further, when the sequence contained oxidized Met, the Met loss from the bare mass was considered as representative of the naked peptide mass. We then used electron-based fragmentation MS2 spectra for localization. Here, all plausible localizations were considered, regardless of search result output. We confirmed the presence of fragment ions in ETD or EThcD that were between potential glycosylation sites, if sufficient c/z ions were present then a glycan mass was considered localized.

Other important considerations during manual validation of search results:

- After the initial identification of a particular peptide sequence in a strong spectrum, less stringent conditions were needed if the same peptide occurred with a different glycan structure. We used the stronger fragmentation spectrum to determine characteristic fragmentation masses, and then weaker spectra were assigned based on fragment abundance similarity (akin to manual spectral matching).

- In EThcD spectra, ions that had the glycosylation present on the fragment (i.e., c/z ions) were considered more important for localization than the ones that show the fragment without glycosylation (i.e., b/y ions).
- For peptides with a 138/144 ratio under 1.2, we assumed that all glycans in the spectrum were core 1 structures. That is, if two sites did not have coverage in ET(hc)D but the glycan composition was N2, H2N2, or H2N2A4, it was assigned as two N1, H1N1, or H1N1A2 structures to both sites. This was not the case if there was an oxonium ion present at 407 m/z, which would indicate the presence of 2 HexNAcs in a single glycan structure.
- When multiple analyses were being compared (e.g. different O-glycoprotease digestions) that could be expected to have the same unique glycopeptides, any given assignment was checked across all files at the same time. Localization in one file was assumed in the other file(s) if the scan was too weak to localize and (a) retention time was identical, and (b) no contradictory information existed in the sequence and/or localization.
- The list of results generated by trainees was reviewed by senior members of the lab. Additionally, any unlikely identifications (e.g., uncommon glycans, a single identification of a glycopeptide) were manually annotated by the trainee and submitted to a senior member for approval before they were included in the curated results.

To obtain as many glycopeptides as possible for the glycoproteomic maps throughout the manuscript, further identifications were made beyond search result verification. This can allow for identification of mutations, uncommon glycan structures, and anything else that did not fall within the search parameters. We performed these analyses in a number of different ways:

- We extracted the HexNAc fingerprint ion (204.0687) from MS2 spectra, which gives overall intensity of the oxonium ion throughout the run. Alternatively, we extracted any HCD scans that triggered an ET(hc)D scan. If any of these highly abundant glycosylated species were not identified by the search algorithm, they were manually de novo sequenced.
- Further, if we did not obtain 100% sequence coverage for a mucin domain, peptide sequences were predicted based on the cleavage motif(s) of the enzyme(s) used. Candidate spectra were found by extracting all scans with oxonium ions and expected fragments from the peptide (e.g., Y0 and N-terminal Pro cleavages). These scans were further filtered by subtracting the naked peptide mass from the precursor mass and matching the remaining mass to different combinations of glycan masses. For spectra that had mass differences matching a glycan mass, the sequences were then validated and localized as described above.
- Finally, to assure we obtained the highest number of glycoforms (i.e., peptides with different glycan compositions and/or glycosites), we used a similar extraction technique. For any potential glycopeptide that was poorly scored/identified by the search algorithm, we extracted expected fragments and any spectra with a new glycan composition were then validated and localized as described above. For example, TIAVFT (TIM-4) had a lot of identified glycoforms (>50) because the HCD was good, whereas the search algorithm only identified one TVRT (TIM1) unique glycopeptide. Using this technique we found

a total of 6 unique glycopeptides from TVRT, all of which were more abundant than the identified one (2xN1), but had HCD spectra (i.e., less b/y fragment ions) because the glycans were more extended.

Abundance calculations

To determine individual glycopeptide abundances, we first generated extracted ion chromatograms of monoisotopic masses and calculated areas under the curve (AUCs). To avoid bias toward smaller glycan structures while minimizing the inconvenience of typing multiple isotope masses into XCalibur QualBrowser (Thermo) for each peptide, abundances were calculated using a variable number of isotopes. Only the ¹²C peak was used for anything under 1600 Da, the ¹³C was also included up to 2400 Da, and three isotopes were collected over 2400 Da. These values were chosen based on the predicted isotopic distributions for given intact peptide values to allow the majority of the signal to be incorporated. All charge states were included when determining the abundance of a given glycoform. Glycopeptide abundances were taken using the file collected with ETD, even if localization took place using the paired EThcD file. If the protein required a secondary file with PNGaseF treatment in order to identify O-glycosites near N-glycosylation sites (TIM-4, TIM-3). In these cases, the PNGaseF treated file was used to record abundances.

To generate figure 1A and determine the most abundant glycan at each site of TIM-3 and TIM-4, we used the above method with additional steps. First, the most abundant core type was determined based on combined area from each category (Tn, core 1, core 2, or “other”), then the most abundant glycan within the category was chosen to investigate. Thereafter, when a residue was observed with a particular glycan structure, but with different glycans throughout the rest of the peptide (i.e., different glycoforms), the abundances of all peptides containing that glycosite with that glycan were summed to get the value for the “most abundant glycan”.

Manual Annotation (“Markups”)

Please see Appendix 1 for representative spectra that have been annotated. These “markups” have a cover sheet showing the sequence, assigned localization, fragment ions identified in the spectra (here, HCD calculated sans glycosylation, ETD with glycosylation), difference from calculated mass, and any additional comments/reasoning/observations.

On the first page, a checkmark indicates that the fragment mass associated with a fragmentation type is present in an included spectrum (b/y for HCD, c/z for ExD). Low-abundance ions were denoted as weak (w) or very weak (vw). Assignments that were ambiguous (e.g., assignable to multiple possible fragment species) were marked with a question mark (?). Neutral losses and fragment ions from the non-standard series were denoted here for each fragmentation type: HCD: water loss (o), ammonia loss (N), a-ion (a), fragment observed with glycosylation attached (*); ETD: a●-ion (a), ETD y-ion (y). EThcD may include a combination of these notations. Internal fragments seen in HCD were shown using a line next to the vertical sequence that covers the relevant amino acid stretch.

Following the cover page is one or more annotated HCD spectra followed by annotated ETD and/or EThcD spectra. Checkmarks on these spectra included the fingerprint ions expected from the assigned glycan structure(s). Sequence-informative ions were labeled with a/b/c/y/z, the fragment number, and the charge. Neutral losses were usually depicted with a line from the originating fragment but may be shown as the fragment assignment and the loss (e.g., b4⁺-H₂O).

The last page of contains more general information that can be helpful for making an assignment. On the left, it displays the total ion current, base peak, elution profile of the anticipated charge state(s), and the MS2 scan times. On the right, zoomed MS1 peaks of the precursor showed the isotopic distribution of the peak, whether there were co-isolated species, and the detected exact mass of the precursor. If multiple peptides with the same overall glycan composition were detected, this page will have labels above the peaks connecting them to their respective spectra, labeled with a number over the peak which is also on the cover page for the assignment. If multiple files were used for an identification, they will be marked in different colors to show where the information was collected.

On page 3, the authors say „To determine SmE’s mucin selectivity, we digested glycoproteins with and without mucin domains. SmE preferentially cleaved the mucin proteins C1 esterase inhibitor (C1-Inh), CD43, and TIM-1, whereas it did not significantly cleave the non-mucin glycoproteins fibronectin and fetuin (Figure S2).” – From Figure S2 it is not evident that CD43 is indeed digested. Please clarify.

This is addressed in the supplementary paragraph “Limitations of SmE for glycoproteomic analyses”:

“Finally, we observed that SmE exhibited reduced proteolytic activity on recombinant proteins expressed in murine-derived cell lines, as exemplified by (a) the digestion of TIM-1 from NS0 and HEK cells over the course of six hours (Figure S10) and (b) limited digestion of CD43 derived from NS0 cells (Figure S2).”

The CD43 used in Figure S2 was expressed in NS0 cells; a sentence clarifying this point was added to the figure caption.

Figure 1B, P1 and P1’ pie chart – what percentage of the peptides were actually glycosylated at their respective termini?

For P1’, all peptides that were definitively cleaved by SmE were glycosylated. Some peptides were generated via trypsin, thus would appear to have a non-glycosylated N-terminus. For P1, it is difficult to assess given that localization on the C-terminus of mucinase-generated glycopeptides is challenging due to the lack of charge normally given by an R/K. Thus, any estimate we could give is likely an underestimate of how often that position is glycosylated.

In the samples that were not treated with trypsin, did the authors identify any non-glycosylated peptides? If yes, please include the respective data. (Please reflect after reassessing the data.)

In the samples that were only treated with SmE, we did not observe any non-specific cleavage from the enzyme.

Which PSMs were included in the pie charts? – the number of glycopeptides listed in the O-Pair outputs (repository) is much higher than 587 (for P1’).

The PSMs included in the pie charts are those derived from the SI Tables with curated results. This is a good example of how the raw search output content (downloaded from the PRIDE repository) will be dramatically

different than the curated files, since we did not agree with a large majority of the O-Pair assignments. Only the manually approved glycopeptides were counted for the pie charts.

It is not clear how exactly the authors assembled the major glycoforms for MD simulation. How were different glycoforms of the same molecular weight treated? These might or might not separate during reversed phase separation. For example, level1 PSMs were assigned to three SmE-generated glycoforms of the SADTVLL peptide of TIM4 with the total H2N2A1 glycan composition: SADT[H2N2A1]VLL, S[H2N2A1]ADTVLL, and S[H1N1]ADT[H1N1A1]VLL. These isoforms (if these are present at all, as all higher-scoring PSMs belong to the S[H1N1]ADT[H1N1A1]VLL isoform) coelute. Moreover, many glycosylation sites might be covered by multiple peptides. How was this issue treated?

We apologize for not being clear on how abundances were calculated for the MD simulations. We have now included a detailed description of manual validation and abundance calculations in the SI Methods (copied above).

Please include source information for Figure S16 (file, scan#). Please also include the respective annotated HCD spectrum.

We thank the reviewer for the suggestion and have updated Figure S16 to the following:

Figure S16. Recombinantly expressed TIM proteins contain complex glycan structures. TIM-1 was subjected to digestion with SmE followed by MS analysis and manual data interpretation. HCD and EThcD spectra from glycopeptide TQLFLEHSLI is depicted above, bearing a core 4 glycan structure. We observed full sequence coverage and unambiguous site-localization. This glycopeptide was observed in the file named 220911_KEM_TIM1_HisTag_SmE_EThcD at the scan numbers indicated in the figure.

Appending an explanatory TOC to the supplementary data, and also to the data in the repository would be extremely useful. For example, what do modification sites highlighted in purple („implied”) in the „unique mod sites” sheet stand for?

We apologize for the lack of clarity and have now included legends for all of the SI tables. Implied cleavage sites are highlighted in purple; if we observed a cleavage event but did not detect the associated C-terminal peptide, we assumed the site was glycosylated since we know the cleavage motifs for all of the enzymes used. This information is also included in the table legends.

What is the difference between different comparison psmtsv files like oglyco_C1 INH_Comparison_1, oglyco_C1-INH_Comparison_2 and oglyco_C1-INH_Comparison_3?

These are different search parameters for different conditions such as fragmentation types (ETD vs. EThcD), digestion conditions (mucinase only vs. mucinase + trypsin), and number of allowed glycosites or missed cleavages. We often use several search parameters to inform our results. The psmtsv files can be downloaded from the repository to distinguish the differences in search parameters.

Why is the „Most common per site” tab empty in Supplementary data 412631_0_data_set_7320029_rpmqy7.xlsx? Please rename the supplementary files to reflect their contents/include a TOC file for all supplementary files.

We thank the reviewer for pointing out this oversight. Since we did not perform MD simulations of TIM-1, we did not calculate the abundances of glycans at each site. This tab has been removed from the file.

Reviewer 3:

Chongsaritsinsuk et al presents a new and powerful technique (a workflow) based on the potential of mucinase SmE to enable a deep characterization of the mucinome of a key family of immune checkpoint inhibitors, the TIM family. The toolkit that was developed is definitely a step forward towards a comprehensive characterization of mucin-domain glycoproteins.

The group has been carrying out an interesting methodological development. in previous studies. focused on the mapping of the mucinome of different mucin-domain glycoproteins, which is a challenging approach given the dense glycosylation characteristic of mucin domains. In this study, they demonstrated that TIM-3 has fewer O-glycosites when compared to TIM-1 and -4, associated with shorter persistence length and higher flexibility in comparison with TIM-4.

The manuscript is pretty much based in the molecular characterization of mucin domain of TIM family highlighting that predicted glycosylation sites in the TIM family (currently described in literature) could have significant limitations as it does not provide a clear insight on the true glycoproteomic landscape of the mucin domain glycoproteins.

We thank the reviewer for their kind words and for the time spent reviewing our manuscript.

Despite this major methodological advance with a broad range of potential applications in other glycoproteins harboring mucin domains, the major concern of this ms is the fact that it is not clear how this differential glycoproteome (mucinome) directly affects the activity and function of TIM molecules:

- What is the impact of this differential mucinome in TIM protein ligand binding and intracellular signaling? What

is the impact of SmE clearance of mucins from TIM3 in its binding affinity to its ligand (e.g. Galectin 9; CEACAM1; HMGB1). What are the functional implications of having fewer O-glycosites found to be present in TIM-3 in comparison with TIM-1 or -4 in terms of interaction with its ligands?

The inclusion of functional assays will definitely improve the potential application of having a clear mucinome characterization in terms of structure-function relationship, opening new avenues for leveraging the glycoproteomic landscape of TIM family to improve cancer immunotherapy.

We appreciate the reviewer's suggestion regarding functional assays and have now included an entirely new section of the manuscript. We previously observed that the affinity of TIM-3 to PtdSer is much lower than that of TIM-4 (Smith *et al.*, *Biochem J*, 2021), and wondered if glycosylation could be contributing to these interactions. To be sure, we found that Gal-9 (a bivalent glycan-binding protein) creates lattices with TIM-3 that increases its effective binding to PtdSer. Given that TIM-4 has a much longer persistence length and protrudes from the glycocalyx, it is likely able to bind PtdSer without requiring this lattice formation. The detailed additions are shown below. Additionally, we added a figure and the requisite methods to the Supplemental Information.

Functional consequences of TIM protein mucin domains

Beyond the differences in structural rigidity and extension imparted by altered density of O-GalNAc glycosylation,^{66,67} we sought to uncover other ways in which variably glycosylation of mucin domains might influence protein function. Indeed, our MD simulations revealed that TIM-4 extended approximately 5-fold further from the cell surface than TIM-3 despite having only 50% more amino acids in the extracellular region. Differences in the protrusion of these receptors from the cell surface should impact how accessible they are to their ligands. The immune cell glycocalyx is dominated by CD43 and CD45, which extend up to 45 and 51 nm from the cell surface, respectively.⁶⁸⁻⁷⁰ With a persistence length of ~41.5 nm, TIM-4 could span much of this distance, bringing its IgV domain close to the external environment. Conversely, the less glycosylated TIM-3 extracellular region relaxes to only ~8.1 nm, suggesting that it would remain obscured by the surrounding glycocalyx. As a result, PtdSer, a ligand for both TIM-3 and TIM-4, might more easily engage TIM-4 on immune cells to alter cell signaling. Interestingly, TIM-3 has multiple ligands, and it has been proposed that these ligands can all interact with TIM-3 simultaneously, thereby potentially altering the binding interactions seen with individual isolated ligands.^{11,54} For instance, the TIM-3 ligand Gal-9 is a bivalent lectin that binds glycoproteins decorated with the LacNAc (Gal-GlcNAc) structure and has been shown to crosslink receptors at the cell surface.^{71,72} We hypothesized that by employing both of its carbohydrate recognition domains (CRDs), Gal-9 can create networks of TIM-3 at the cell surface. Further, these lattices could result in patches of more accessible TIM-3, thereby increasing affinity (and avidity) for other ligands like PtdSer.

Figure 5. Gal-9 crosslinks TIM-3 to enhance affinity to PtdSer. (A) Gal-9 and the extracellular region of TIM-3 were incubated in the presence of the crosslinking reagent disuccinimidyl suberate (DSS). Western blotting for TIM-3 indicated that higher molecular weight species formed with increasing concentration of Gal-9. (B) Surface plasmon resonance was used to quantify the impact of (bivalent) Gal-9^{WT}, with those of monovalent Gal-9^{R200D}, or Gal-9^{R65D} variants on the binding of TIM-3 to immobilized membranes containing 20% PtdSer and 80% phosphatidylcholine. Lipid vesicles were immobilized on an L1 sensorchip, and protein samples were flown over the surface with 1 mM CaCl₂. The increase in response units (RU) observed for TIM-3 in the presence of Gal-9^{WT}, Gal-9^{R65D}, or Gal-9^{R200D} is plotted as a function of Gal-9 concentration. (C) Sensorgrams of TIM-3, TIM-3 and Gal-9^{WT}, TIM-3 and Gal-9^{R65D}, and TIM-3 and Gal-9^{R200D}. The black triangles below the x-axis indicate the start of the sample injection (~75 s), the end of the sample injection (~375 s), and the regeneration (~510 s). Data points from the dissociation phase, (~375 s – 500 s), were fit using a nonlinear regression model that was used to estimate k_{off} values. (D) Bar graph showing the calculated k_{off} values for the various conditions tested. Significance was tested using one-way ANOVA in Graphpad PRISM **** indicates p value < 0.0001.

To test this hypothesis, we assessed the ability of Gal-9 to crosslink TIM-3 by incubating the bivalent lectin with the complete extracellular region of TIM-3 in the presence of the chemical crosslinking reagent disuccinimidyl suberate (DSS). For these experiments, we employed the variant of Gal-9 developed by Itoh and colleagues to enhance stability and solubility of the recombinantly expressed protein.⁷³ Only when TIM-3 and Gal-9 were incubated in the presence of DSS were larger molecular weight complexes observed above the band for monomeric TIM-3 (Figure 5A, lanes 2-4, Figure S19). As the concentration of Gal-9 increased the bands these higher molecular weight species intensified, indicating that the shift of TIM-3 into complexes was dependent on the amount of Gal-9 present. Furthermore, the ladder pattern of TIM-3 detected suggests formation of a lattice including these two binding partners.

We next asked whether oligomerization of TIM-3 by Gal-9 might enhance its avidity for PtdSer in cell membranes. Our previous findings demonstrated that TIM-3 binds PtdSer-containing membranes, and that mutations that impair TIM-3 binding to PtdSer reduce its impact on T cell signaling.⁷⁴ Using surface plasmon resonance (SPR), we quantified the TIM-3 binding to PtdSer-containing membranes in the presence and absence of wildtype Gal-9 (Gal-9^{WT}) or variants of Gal-9 with mutations in one CRD (Gal-9^{R65D} or Gal-9^{R200D}) that render the protein monovalent – which should reduce crosslinking of TIM-3. Adding Gal-9^{WT} to TIM-3 enhanced its binding to PtdSer-containing surfaces in a dose-dependent manner (**Figure 5B**), whereas the mutated Gal-9 variants did not, indicating that the increased PtdSer binding affinity reflects formation of a TIM-3/Gal-9 lattice.

We were not able to quantify the effects of Gal-9 induced crosslinking on the affinity of TIM-3 for PtdSer because of problems with Gal-9 precipitation. We therefore exploited the SPR approach to perform kinetic analysis and calculate k_{off} (the rate of dissociation) from PtdSer-containing membranes for the different TIM-3/Gal-9 complexes. Following rapid association with the surface, equilibrium binding was reached for all samples except TIM-3 with Gal-9^{WT} within 100-350 s (**Figure 5C**). The continued escalation of signal for the TIM-3/Gal-9^{WT} sample likely reflects continued growth of a TIM-3/Gal-9 lattice on the PtdSer-containing membranes immobilized on the sensorchip. Moreover, Gal-9^{WT} significantly slowed the dissociation of TIM-3 from the PtdSer-containing surface when compared with TIM-3 alone or TIM-3 bound to monovalent Gal-9 variants (**Figure 5C**). The resulting curves were fit with a one-phase exponential decay model to estimate k_{off} and quantify the effect of Gal-9^{WT} on TIM-3/PtdSer binding (**Figure 5D**). We found that k_{off} for TIM-3 in the presence of Gal-9^{WT} ($0.030 \pm 0.002 \text{ sec}^{-1}$) was ~9-fold slower than k_{off} for TIM-3 alone ($0.27 \pm 0.03 \text{ sec}^{-1}$). Mutation either of the two CRDs in Gal-9 abrogated this effect, with k_{off} for TIM-3 with the mutant Gal-9 variants being unchanged ($0.26 \pm 0.01 \text{ sec}^{-1}$ for Gal-9^{R65D} and $0.22 \pm 0.01 \text{ sec}^{-1}$ for Gal-9^{R200D}). These data argue that Gal-9^{WT} significantly enhances the avidity of TIM-3 for PtdSer through crosslinking multiple TIM-3 molecules and enhancing their access to PtdSer on an opposing membrane.

Although Gal-9 is known to interact with other immune modulators,⁷⁵⁻⁷⁹ no experimental evidence suggests Gal-9 binds to TIM-4. TIM-4 has a substantially higher binding affinity for PtdSer than that of TIM-3⁷⁴, and also extends further from the glycocalyx than TIM-3 as a result of its increased density of glycosylation, according to our analysis. Thus, TIM-4 likely does not require lattice formation to increase local receptor concentration and improve binding strength. While these findings are preliminary in nature, they shed new potential light on how mucin-type glycosylation impacts the structure and function of important cellular receptors. In the cases of TIM-3 and TIM-4, we suggest that the relatively sparse glycosylation allows bivalent lectin binding to control avidity of an otherwise low-affinity PtdSer binder for immune cell control. For TIM-4, which has a high-affinity PtdSer-binding site, dense glycosylation can project it from the glycocalyx in a way that allows it constitutively to bind PtdSer, reflecting the differences in TIM-3 and TIM-4 function.

Appendix 1: Manual glycopeptide annotation

The following appendix includes representative spectral annotations. Each glycopeptide will have the following associated with it:

- A cover sheet (1 page) showing the sequence, assigned localization, fragment ions identified in the spectra (here, HCD calculated sans glycosylation, ETD with glycosylation), difference from calculated mass, and any additional comments/reasoning/observations. On this page, a checkmark indicates that the fragment mass associated with a fragmentation type is present in an included spectrum (b/y for HCD, c/z for ExD). Low-abundance ions were denoted as weak (w) or very weak (vw). Assignments that were ambiguous (e.g., assignable to multiple possible fragment species) were marked with a question mark (?). Neutral losses and fragment ions from the non-standard series were denoted here for each fragmentation type: HCD: water loss (o), ammonia loss (N), a-ion (a), fragment observed with glycosylation attached (*); ETD: a●-ion (a), ETD y-ion (y). EThcD may include a combination of these notations. Internal fragments seen in HCD were shown using a line next to the vertical sequence that covers the relevant amino acid stretch.
- One or more annotated HCD spectra followed by annotated ETD and/or EThcD spectra (2-4 pages). Checkmarks on these spectra included the fingerprint ions expected from the assigned glycan structure(s). Sequence-informative ions were labeled with a/b/c/y/z, the fragment number, and the charge. Neutral losses were usually depicted with a line from the originating fragment but may be shown as the fragment assignment and the loss (e.g., b4+-H₂O).
- More general information that can be helpful for making an assignment (1 page). On the left, it displays the total ion current, base peak, elution profile of the anticipated charge state(s), and the MS2 scan times. On the right, zoomed MS1 peaks of the precursor showed the isotopic distribution of the peak, whether there were co-isolated species, and the detected exact mass of the precursor. If multiple peptides with the same overall glycan composition were detected, this page will have labels above the peaks connecting them to their respective spectra, labeled with a number over the peak which is also on the cover page for the assignment. If multiple files were used for an identification, they will be marked in different colors to show where the information was collected.

REVIEWERS' COMMENTS

Reviewer #2 (Remarks to the Author):

Chongsaritsinsuk and coworkers addressed all my questions and the additional supplemental methods help the reader to better understand the data interpretation workflow.

I only have a minor remark regarding the revised SFigure 16 – I think that this is a somewhat ambiguous assignment. The HCD spectrum indicates a direct linkage of HexNAcHex to the peptide, and the 2*HexNAc loss along with the abundant HexNAc2 oxonium ion (m/z 407) indicate a terminal HexNAc2 disaccharide. These ions are not compatible with the proposed glycan structure (green border on below figure) but fit an alternative core-2 LacdiNAC-like structure (red border). This latter glycan was identified in human gastric mucins (Jin MCP 2017). On the other hand, the ion representing the N1H1A1 loss (m/z 929(2+)) in EThcD is not compatible with this alternative glycan but can be assigned to the proposed core-4 structure. The complete c ion series proves that a single glycan modifies the peptide, therefore my best guess is that the spectrum represents a pair of coeluting isomeric glycoform.

As the presence of any core-4 glycopeptide is not a major highlight of the findings, I recommend to remove these data from the manuscript.

Reviewer #3 (Remarks to the Author):

The authors did an amazing job in carefully answering all the comments by providing new and detailed experiments that clearly clarify the impact of TIM mucinome in protein function/activity, namely in Gal9 and PtdSer interactions. The new data provided clearly strengthened the content and novelty of the study that is now worth published.

Response to Reviewers

Reviewer #2:

Chongsaritsinsuk and coworkers addressed all my questions and the additional supplemental methods help the reader to better understand the data interpretation workflow.

We thank the reviewer for their time spent handling our manuscript.

I only have a minor remark regarding the revised SFigure 16 – I think that this is a somewhat ambiguous assignment. The HCD spectrum indicates a direct linkage of HexNAcHex to the peptide, and the 2*HexNAc loss along with the abundant HexNAc2 oxonium ion (m/z 407) indicate a terminal HexNAc2 disaccharide. These ions are not compatible with the proposed glycan structure (green border on below figure) but fit an alternative core-2 LacdiNAc-like structure (red border). This latter glycan was identified in human gastric mucins (Jin MCP 2017). On the other hand, the ion representing the N1H1A1 loss (m/z 929(2+)) in EThcD is not compatible with this alternative glycan but can be assigned to the proposed core-4 structure. The complete c ion series proves that a single glycan modifies the peptide, therefore my best guess is that the spectrum represents a pair of coeluting isomeric glycoform.

As the presence of any core-4 glycopeptide is not a major highlight of the findings, I recommend to remove these data from the manuscript. (I attach a figure to illustrate my point.)

We have removed SI Fig 16 to accommodate the reviewer's concern.

Reviewer #3:

The authors did an amazing job in carefully answering all the comments by providing new and detailed experiments that clearly clarify the impact of TIM mucinome in protein function/activity, namely in Gal9 and PtdSer interactions. The new data provided clearly strengthened the content and novelty of the study that is now worth published.

We thank the reviewer for their kind words and for the suggestion to add functional data to our study. The manuscript is stronger because of these studies.